# KAT5 regulates neurodevelopmental states associated with G0-like populations in glioblastoma

Anca B. Mihalas[1], Sonali Arora [1], Samantha A. O'Connor [2], Heather M. Feldman[1], Christine E. Cucinotta[3,4], Kelly Mitchell[1], John Bassett[5], Dayoung Kim[4], Kang Jin [6], Pia Hoellerbauer[1], Jennifer Delegard[7], Melissa Ling[8], Wesley Jenkins [8], Megan Kufeld[1], Philip Corrin[1], Lucas Carter[1], Toshio Tsukiyama[4], Bruce Aronow [6], Christopher L. Plaisier [2], Anoop P. Patel [9,10,11] ✉ & Patrick J. Paddison [1,8] ✉

Quiescence cancer stem-like cells may play key roles in promoting tumor cell heterogeneity and recurrence for many tumors, including glioblastoma (GBM). Here we show that the protein acetyltransferase KAT5 is a key regulator of transcriptional, epigenetic, and proliferative heterogeneity impacting transitions into G0-like states in GBM. KAT5 activity suppresses the emergence of quiescent subpopulations with neurodevelopmental progenitor characteristics, while promoting GBM stem-like cell (GSC) self-renewal through coordinately regulating E2F- and MYC- transcriptional networks with protein translation. KAT5 inactivation significantly decreases tumor progression and invasive behavior while increasing survival after standard of care. Further, increasing *MYC* expression in human neural stem cells stimulates KAT5 activity and protein translation, as well as confers sensitivity to homoharringtonine, to similar levels to those found in GSCs and high-grade gliomas. These results suggest that the dynamic behavior of KAT5 plays key roles in G0 ingress/egress, adoption of quasi-neurodevelopmental states, and aggressive tumor growth in gliomas.

It is now well established from single cell RNA-seq studies that glioblastoma (GBM) tumors are complex, yet maligned, neurodevelopmental ecosystems, harboring diverse tumor cell types, including cells resembling astrocytes, neural progenitors, oligodendrocyte progenitor cells, mesenchymal cells and radial glial cells, all of which presumably contribute to tumor growth and homeostasis in specific ways (e.g.[1,2],). Rather than deterministic hierarchies, this cellular and developmental heterogeneity may be better explained by phenotypic plasticity[3], where cells can transition from one state to another through aberrant development paths not observed in healthy tissues.

However, single cell data sets have failed to produce general models for transitions in and out of specific developmental and

[1]Human Biology Division, Fred Hutchinson Cancer Center, Seattle, WA 98109, USA. [2]School of Biological and Health Systems Engineering, Arizona State University, Tempe, AZ 85281, USA. [3]College of Arts and Sciences, Department of Molecular Genetics, Ohio State University, Columbus, OH 43210, USA. [4]Basic Sciences Division, Fred Hutchinson Cancer Center, Seattle, WA 98109, USA. [5]Department of Medicine, Karolinska Institute, Huddinge, Sweden. [6]Division of Biomedical Informatics, Cincinnati Children's Hospital Medical Center, Cincinnati, OH 45229, USA. [7]Department of Neurosurgery, University of Washington, Seattle, WA 98195, USA. [8]Molecular and Cellular Biology Program, University of Washington, Seattle, WA 98195, USA. [9]Department of Neurosurgery, Duke University, Durham, NC 27710, USA. [10]Preston Robert Tisch Brain Tumor Center, Duke University, Durham, NC 27710, USA. [11]Center for Advanced Genomic Technologies, Duke University, Durham, NC 27710, USA. ✉e-mail: anoop.patel@duke.edu; paddison@fredhutch.org

proliferative states in tumors. One reason is that, for GBM tumor cells, there are no pre-existing *universal* markers that neatly resolve subpopulations into quiescent, "primed", G1, or differentiated cellular states[4,5] as is the case, for example, for adult mammalian neurogenesis[6,7].

From histopathology studies, it is estimated that ~30% of glioblastoma tumor cells at a given moment are actively dividing (e.g., stain positive for the proliferation marker Ki67)[8]. The phenotypic behavior of the remaining ~70% of cells is largely unknown; although it is presumed that some portion are in a state of transient or long-term quiescence[4,5]. The interplay between pathways and signals promoting cell cycle ingress and those governing its egress likely play key roles in generating cellular heterogeneity observed in GBM as well as its resistance to chemoradiation. Here, we sought to identify key regulators of G0-like states in human GBM stem-like cells (GSCs).

## Results

### A functional genomic screen to identify genes that regulate G0-like states in human GSCs

We used patient-derived GBM stem-like cells (GSCs) to identify genes with roles in regulating G0-like states. GSCs are isolated and cultured in serum-free conditions directly from GBM tumors that allow retention of the development potential, gene expression patterns, and genetic alterations found in the patient's tumor[9–12]. In these conditions, GSCs typically have shorter overall transit times between mitosis and the next S-phase (i.e., G0/G1) when compared to hNSCs. G0/G1 is ~12 h for cultured GSCs versus ~33 h for hNSCs[5]. This large difference is due to the additional time cultured NSCs spend in a transient G0 state[5], which is opposed by oncogenic lesions in GBM cells that increase the likelihood of cell cycle entry (e.g., Rb-axis alterations)[13]. We sought to identity genes which when inhibited could trigger a pronounced G0-like state in GSCs. For the functional genomic screen, we used IDH1[wt] proneural GSC isolate 0827, which was derived from a surgically resected recurrent tumor specimen of an adult GBM patient who had received standard of care (radiation and temozolomide)(Dr. H. Fine, personal communication; confirmed by Dr. PJ Cimino (NIH/NINDS) via NIH's tissue bank database).

This isolate was used in conjunction with a fluorescent protein G0 reporter system consisting of a G1/G0 phase mCherry-CDT1 reporter[14] combined with a p27-mVenus G0 reporter[15,16]. Both CDT1 and p27 are targeted for proteolysis by the SCF[Skp2] E3 ubiquitin ligase complex in S/G2/M phases of the cell cycle but not in G1 or G0[17]. p27 is additionally regulated in G1 via targeted degradation by the Kip1 ubiquitylation-promoting complex at the G0-G1 transition[17]. As a result, p27 only accumulates during G0, while CDT1 is observed in both G0 and G1. The p27 reporter was constructed with a p27 allele that harbors two amino acid substitutions (F62A and F64A) that block binding to Cyclin/CDK complexes (preventing functional activity) but do not interfere with its cell cycle-dependent proteolysis (Oki et al. 2014) (Supplementary Fig. 1a).

To confirm the reporter system, we assayed steady-state EdU incorporation, an indicator of cells actively replicating DNA, in GSC-0827 reporter cells (Supplementary Fig. 1b). Consistent with reporting of G0-like states in GSCs, EdU incorporation is significantly suppressed in p27-mVenus[+] and especially p27-mVenus[hi] (p27[hi]) cells. Further, to ensure that p27[hi] cells could re-enter the cell cycle, p27[hi] cells were sorted, recultured, and assayed 7 days later for EdU incorporation and p27-mVenus levels. The p27hi cells showed high, but somewhat diminished, EdU incorporation rate of 30% versus 35% for control cells, and also had higher residual p27-mVenus expression. These results are consistent with a preponderance of p27[hi] cells being division capable and entering the cell cycle with delayed and somewhat variable kinetics as cells exit from G0 (Supplementary Fig. 1c).

To identify key regulators of G0 ingress/egress we used GSC-0827 p27 reporter cells in a genome-wide knockout screen to identify genes

which when inhibited trap cells in G0-like cells (Fig. 1a). Comparing sgRNAs in unsorted versus sorted double positive populations yielded 75 genes enriched and 37 depleted in p27[hi]CDT1+ cells (FDR < .01) (Fig. 1b; Supplementary Fig. 1d–f). The most prominent enriched gene sets included those involved in ribosome assembly and ribosome protein coding genes (e.g., *RPL5*, *RPS16*, etc.) and the Tip60/NuA4 lysine acetyltransferase complex (e.g., *ACTL6A*, *EP400*, *KAT5*, *TRAPP*), with the KAT5 catalytic subunit scoring among top ten screen hits (Fig. 1b; Supplementary Fig. 1g; Supplementary Data 1).

Among sgRNA targets significantly depleted in p27[hi]CDT1+ cells, we find *FBXO42* and *CDKN1A/p21* were top scoring. *FBXO42* encodes an E3 ubiquitin ligase required for the G2/M transition in GSC-0827 cells[18] and when knocked out in GSC-0827 cells evokes a G2/M arrest, which would deplete G0 cells at steady state levels. CDKN1A/p21 is a cyclin/CDK2 inhibitor that has been shown to promote entry into G0-like states in cultured cells[19], consistent with the screen results. In addition, we find genes associated with the TENTB-ZCCHC14 complex including *PAPD5* and *ZCCHC14*, which help destabilize rRNAs and other non-protein coding RNAs, including miRNAs and TERC[20–22].

Retests of several enriched and depleted G0-trap screen genes largely recapitulated the screen data (Supplementary Fig. 2a, b). KO of *RPS16*, *RPL5*, and KAT5 complex members, resulted in significant increases in p27-mVenus[hi] subpopulations and loss of proliferation in GSC-0827 cells, while KO of *CDKN1A* and *PAPD5* decreased steady-state p27 levels and increased proliferation (Supplementary Fig. 2b, c). However, *RPS16* and *RPL5* KO resulted in significant cell toxicity, while *KAT5* KO cell numbers appeared to be stable even for extended periods (e.g., >2 weeks) (see below).

That genes involved in ribosome function would score as G0 trap mutants is supported by the notion that down regulation of protein synthesis and ribosome assembly are hallmarks of quiescent cells[23,24]; and, conversely, that their activity increases as, for example, neural stem cells transition out of quiescence into an activated state that precedes cell cycle entry[25].

The KAT5/NuA4 lysine acetyltransferase complex targets both histones (H2A variants, H3, and H4) and non-histone proteins for acetylation and functions as a transcriptional co-activator, whose activities are coordinated with multiple transcription factors[26–28]. In addition, the NuA4 complex also participates in DNA double-strand break repair by facilitating chromatin opening[29,30]. However, little is known about the role of KAT5/NuA4 complex activity in GBM biology. Therefore, we chose to further pursue the question of whether KAT5 activity affects transitions in and out of G0-like states in GBM cells.

### *KAT5* knockout triggers a G0-like state in in vitro cultured human GSCs

We examined multiple phenotypes associated with G0-like states using the nucleofection KO assay. First, we determined whether *KAT5* KO could induce a population of cells with G1 DNA content and low total RNA content, a classic indicator of quiescent cells. Five days after introduction of sgKAT5:Cas9 complexes, a significant G1/G0 RNA[low] population emerges in GSC-0827 cells (Fig. 1c, d). Analysis of cell cycle proportions via FUCCI factors (Supplementary Fig. 2e, f) and DNA synthesis rate via EdU incorporation (Fig. 1e; Supplementary Fig. 3a, b) were also consistent with this result. In each case, *KAT5* KO significantly altered these measures in GSC-0827 cells, reducing the frequency of Geminin[hi] and EdU[hi] cells, respectively. Similar results were also observed in five additional human GSC isolates (Supplementary Fig. 3b) and also showed upregulation of p27 (Supplementary Fig. 3c).

Another key hallmark of G0-like states is reversibility, which distinguishes quiescence from differentiation and senescence. Adding back KAT5, 7 days after *KAT5* KO caused the same rapid accumulation of cells as parental controls and the return of parental ratios of cell cycle phases (Supplementary Fig. 3d–f). Taken together, above results

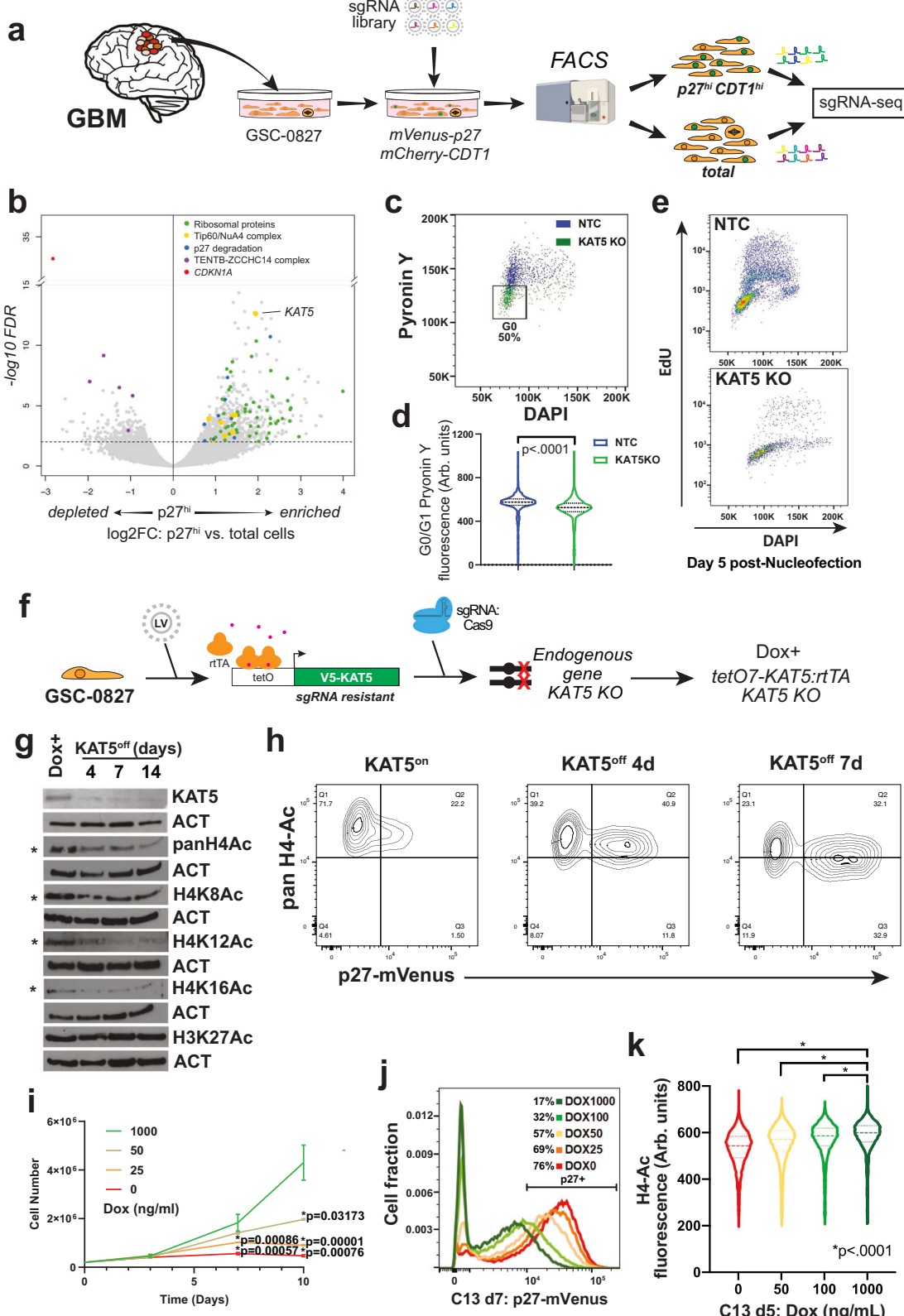

are consistent with KAT5 activity controlling reversible ingress/egress into G0-like states in GSCs.

## Effects of attenuating KAT5 activity in GSC-0827 cells

To better control the dynamics of KAT5 activity, GSC-0827 cells were engineered to have a doxycycline controllable *KAT5* open reading frame and knockout insertion-deletion mutations in the endogenous

*KAT5* gene (Fig. 1f) ("Methods"). We used one such clone, #13 (C13), which demonstrated key characteristics including Dox-dependent KAT5 expression, requirement of Dox+ for continued growth, and competent tumor initiation potential in the presence of Dox.

We first examined inducibility of the *KAT5* ORF and the effects on loss of KAT5 activity by Western blot after Dox withdrawal for 4, 7, and 14 days in C13 cells (Fig. 1g). Among KAT5's key substrates is histone

**Fig. 1 | Identification of *KAT5* as a G0-trap in GSC-0827 cells. a** Schematic of the G0-trap screen. For the screen, GSC-0827 cells containing mCherry-CDT1 (G0/G1) and p27-mVenus (G0) were transduced with a genome-wide CRISPR-Cas9 library, allowed to expand for 10 days, and sorted for double positive cells using the top 20% of p27-mVenus+ cells as a cut off. FACS machine cartoon taken from BioRender. Paddison, P. (2025) https://BioRender.com/o2c4y3l. **b** Results G0-trap screen from sgRNA-seq of p27hi vs. total cell population ($n = 3$; edgeR was used to assess p values and logFC cuttoffs) (Supplementary Data 1). Supporting QC data and gene set enrichment can be found in Supplementary Fig. 1. **c** FACS-based assessment of total RNA and DNA content, using pyronin Y and DAPI, respectively, in GSC-0827 cells via nucleofection of sgRNA:Cas9 RNPs. Additional retest assays are available in Supplementary Figs. 2, 3. **d** Quantification of (**c**) (≥2350 cells per condition from two independent treatments; KS test, $p < .0001$; median and quartiles shown). **e** FACS-based assessment of EdU incorporation after *KAT5* KO using nucleofection of sgRNA:Cas9 RNPs (5 days post-nucleofection) in GSC-0827

cells. Quantification is shown in Supplementary Fig. 3a. **f** GSC-0827 cells were engineered to have a doxycycline controllable *KAT5* open reading frame and knockout insertion-deletion mutations in the endogenous KAT5 gene. We found one such clone, #13 (C13), to have Dox-dependent *KAT5* expression and requirement of Dox+ for continued growth. **g** Western blot analysis of KAT5 and histones H3 and H4 acetylation and methylation status after Dox withdrawal of 0, 4, 7, and 14 days in C13 cells. *indicates KAT5 targets. **h** FACS analysis of histone H4 acetylation and p27-mVenus levels after 0, 7, and 14 days of Dox withdrawal in C13 cells. **i** A growth curve with C13 cells grown in various concentrations of Dox ($n = 3$; mean ± SEM is shown, significance determined by unpaired, 2 tailed student's *t* test). **j** FACS analysis of p27 induction in C13 cells grown in shown concentrations of Dox for 7 days. **k** Violin plot of H4-Ac levels in C13 cells grown in shown concentrations of Dox for 5 days. ≥5703 cells plotted for single trial. KS test was used to test significance, $p < 0.0001$. Median and quartiles displayed. Source data with exact $p$ values are provided with this paper as a Source Data file.

H4. In vitro, KAT5 acetylates H4 amino-terminal tail at four lysine (K) residues (5, 8, 12, and 16) without seeming preference, mainly as monoacetylations[31]. Consistent with these biochemical activities, in C13 cells KAT5 activity is necessary for maintaining "pan" H4-acetylation (ac) levels and also monoacetylations at K8, K12, and K16 (K5 was not examined) (Fig. 1g). However, loss of KAT5 activity did not globally affect other histone marks, including H4K20me3 (associated with heterochromatin[32]), H3K4me2 and H3K27ac (associated with active enhancers and genes[33]), and H3K27me3 (associated with transcriptionally repressed chromatin[34,35]) or histone levels in general (Fig. 1g).

To further assess *KAT5* inhibition in C13 cells, we performed flow analysis 4 and 7 days after Dox withdrawal for intracellular H4-ac marks and p27 expression (Fig. 1h). The results reveal that in Dox+ cells, KAT5 activity and G0-like states are dynamic: H4-achi cells have low p27 levels, while the highest expressing p27 cells have low H4-ac levels. Upon loss of KAT5 activity after Dox withdrawal, we observe progressive loss of H4-ac levels and accumulation of cells with higher p27 levels than in Dox+ control cells. Loss of KAT5 activity in C13 cells also results in reduced rRNA levels, translation rates, and metabolic output (see below), consistent with induction of a G0-like state.

### Single cell gene expression analysis of KAT5 inhibition in GSC cells in vitro

We next investigated cellular states induced by *KAT5* inhibition in cultured GSCs by performing scRNA-seq analysis in GSC-0827 cells (Fig. 2; Supplementary Fig. 5a–c). By this analysis, control GSC-0827 cultures have an embryonic stem cell-like cell cycle, with ~80% of cells in three cycling cell clusters: M/G1 (cluster 2), S/G2 (cluster 0), and G2/M (cluster 5) (Fig. 2a–d).

However, *KAT5* KO causes the cell cycle clusters and cell cycle gene expression to collapse and two new clusters to emerge (clusters 1 and 3). These new states have significant increases in "Neural G0" gene expression[5] (Fig. 2e, f), a computationally defined G0-like state that contains a mixture of genes expressed in adult quiescent NSCs, fetal radial glial (RG) cells, and oligodendrocyte progenitor (OPC) cells, along with absence of cell cycle genes (e.g., *CCNB1* (Fig. 2f)). Neural G0 includes many genes known to participate in neurogenesis or glioma biology: including: *CLU*, a secreted antiapoptotic factor[36]; *F3*, a marker of quiescent GBM cells[37]; *PTN* and its target *PTPRZ1*, which promote stemness, signaling, and proliferation of neural progenitors and glioma tumor cells[38–41]; *S100B*, a chemoattractant for tumor-associated macrophages in glioma[42,43]; and *SPARC* and its ortholog *SPARCL1*, which promote brain tumor invasion and survival[44–46]; and *TTYH1*, required for NSC stemness and Notch activation[47,48] (Fig. 2g, h). Western blot analysis confirmed up regulation of many of these genes after *KAT5* inhibition in GSC-0827 cells, along with down regulation of cell cycle-associated proteins AURKA and MYC and stabilization of endogenous p27 (Fig. 2i). F3 upregulation was confirmed by FACS analysis (Supplementary Fig. 5d).

In addition, Cluster 3 genes up regulated show enrichment for interactions with the extracellular matrix, integrin signaling, and cell migration (Supplementary Data 2-3), which includes genes that participate in the proneural-to-mesenchymal transition in GBM tumors (e.g., *CD44*, *CDH2/N-Cadherin*, *SERPINE1*, *SPP1/Osteopontin*)[49] (However, this was not observed in vivo—see below).

To determine whether *KAT5* KO has similar effects in other patient isolates, we performed scRNA-seq analysis in 8 additional GSC patient isolates and compared *KAT5* KO-specific versus Control *CD8* KO-specific changes in gene expression (Fig. 3; Supplementary Data 6). This comparison required analysis of gene expression changes across 73 separate de novo UMAP clusters as shown in Fig. 3a, which designated clusters as Novel *KAT5* KO, Control *CD8* KO, or Mixed KO (Fig. 3a; Supplementary Fig. 6).

From this analysis, we observed two general trends. First, genes whose expression was preferentially observed in Control *CD8* KO cells tended to be highly enriched for cell cycle genes, a lead indicator of cell proliferation (Fig. 3b, c). Second, genes whose expression was preferentially induced in *KAT5* KO cells were significantly enriched for neural developmental and GBM stem cell genes and transcription factors: including NPAS3, required for stemness of radial glial cells[50], SOX9 and ID4, both with roles in maintaining GBM stem-like cells[51,52] (Fig. 3b, c). In addition, consistent with data from Fig. 2, *KAT5* KO cluster genes showed significant overlap with Neural G0 and radial glial genes, including genes that are likely expressed in quiescent NSC and GBM cells[5] (Fig. 3d).

Because *KAT5* KO appeared to induce *novel* clusters with higher neurodevelopmental and lower cell cycle gene expression, we wondered if these clusters might represent physiological states in tumors that were not apparent in vitro. As a result, we performed scRNA-seq analysis of tumors derived from GSC-0827 and GSC-464T isolates (Fig. 4a, b; Supplementary Figs. 8–11) and then mapped in vitro *KAT5* and Control KO cells from scRNA-seq data onto their respective in vivo tumor references (Fig. 4c–f). The GSC tumors showed more diverse proliferative and non-proliferative states than in vitro cultures did, which were assessed by examining gene expression markers and modules (Fig. 4g, h; Supplementary Fig. 8, 1). For example, GSC-0827 tumors have ~45% of cells in M/G1-S/G2-G2/M clusters compared to 80% for in vitro cultures and new G0/G1 subpopulations with radial glial/astrocytic, OPC, mesenchymal, and hypoxia-associated gene expression. Using the ccAF classifier we observe Neural G0 cells concentrated in the RG/Ac/OPC nondividing populations. These populations had the lowest expression of cell cycle genes, such as *CCNB1* (Fig. 4i, j) and also contain endpoints of mRNA velocity lines, suggestive of cell cycle exist. *KAT5* KO cells showed increased mapping onto Neural G0 clusters furthest away from dividing cell populations for both tumors (Fig. 4k, l). This is consistent with the notion that *KAT5* inhibition causes in vitro cultures to trigger quiescent states normally observed in tumors.

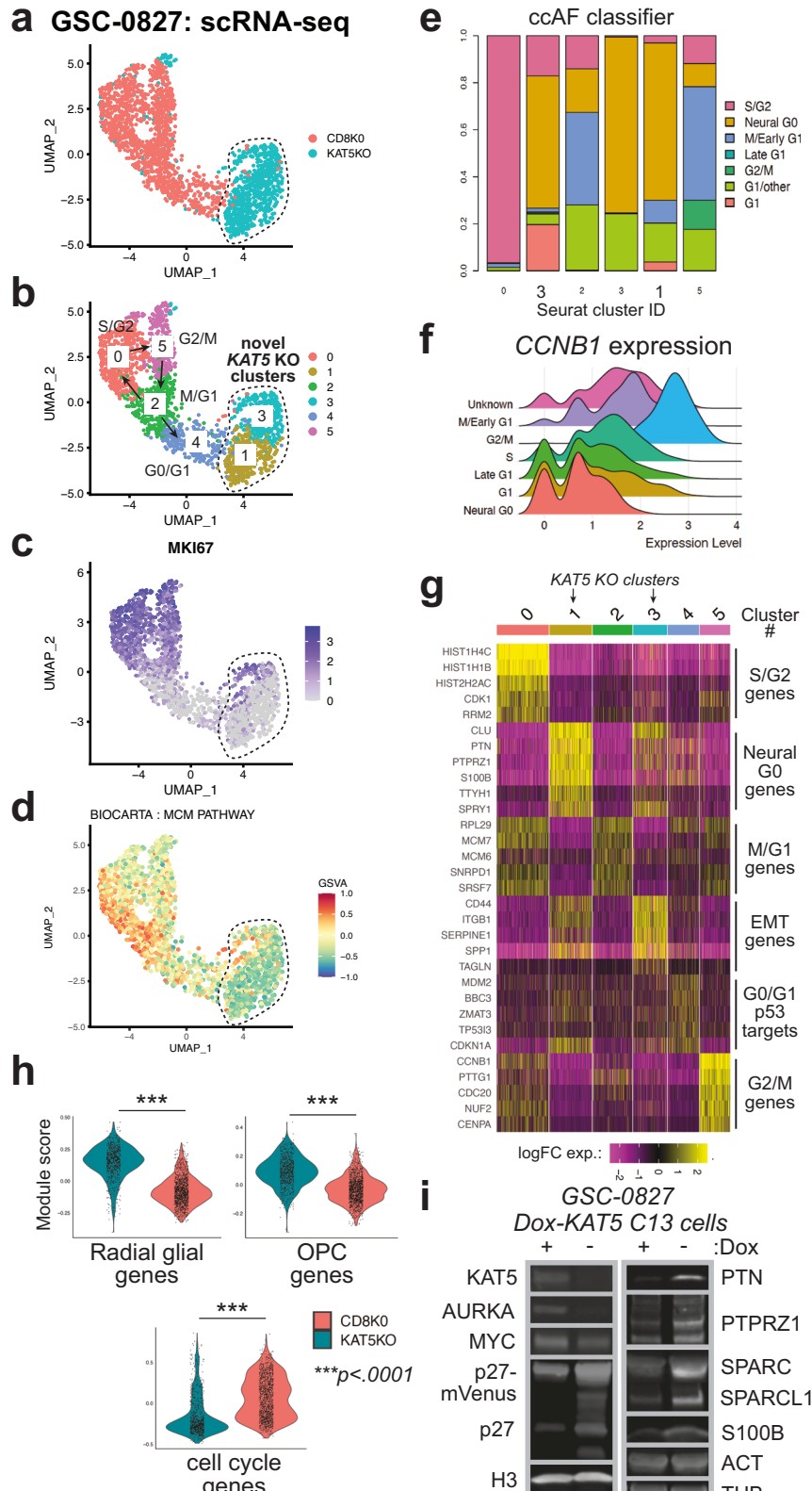

## KAT5 target networks and gene expression and epigenetic alterations induced by *KAT5* loss in GSC-0827 tumors

The results above led us to focus on in vivo assessment of KAT5 activities, including chromatin interactions and maintenance of cell and epigenetic states. To this end, we used C13-derived tumors, where mice were administered Dox in their drinking water to support engraftment and the initial phase of tumor cell growth. After

confirmation of tumor formation by MRI, Dox was either maintained (KAT5[on]) or removed for 6 days (KAT5[off]), after which tumors are harvested and assays performed.

KAT5 chromatin binding was assessed using CUT&TAG[53] analysis for KAT5[on] and KAT5[off] tumor samples (Fig. 5a). These data revealed that KAT5 binds near transcriptional start sites of target genes critical for tumor growth and survival (Fig. 5a, b). ~22% of KAT5-bound genes

**Fig. 2 | Single cell gene expression analysis of GSC-0827 cells after *KAT5* inhibition in vitro. a** UMAP projection of scRNA-seq data for sgCD8 (control) and sgKAT5 cells in GSC-0827 cells 5 days post nucleofection. Filtering scheme and data quality assessment for this data is available in Supplementary Fig. 4. **b** The UMAP projection from **a** showing de novo clusters generated. Cell cycle state predictions using the ccSeurat and ccAF classifiers and RNA velocity analysis for this data from are available in Supplementary Fig. 5. Associated data files include: cluster-based gene expression analysis (Supplementary Data 2) and gene set enrichment for top 200 expressed and top 200 depleted genes for each cluster (Supplementary Data 3, 4, respectively). For control cells, there is one G0-like cluster (cluster 4) which showed some weak expression for p53-associated target genes, suggestive of a DNA damage-induced G0-like state, a common feature of cultured cells (Arora et al., 2017; Spencer et al., 2013) also to some degree Neural G0 genes (see text). This state likely represents the p27hi Edu- population observed in vitro in Supplementary Fig. 1b, which are capable of re-entering the cell cycle. **c** MKI67/Ki67 gene expression, which is only expressed in cycling cells, among cells and clusters from (**b**).

**d** Gene Set Variation Analysis (GSVA) associated with clusters and cell cycle phases showing MCM pathway required for initiation of DNA replication in early S-phase. Cell cycle pathways are shown in Supplementary Fig. 5. **e** Cell cycle phase predictions using the ccAF classifier for scRNA-seq data from (**a**). **f** Analysis of *CCNB1* gene expression in ccAF predicted cell cycle phases: Neural G0 contains fewest cells expressing *CCNB1*, while G2/M the most. **g** Heatmap of representative genes upregulated in scRNA-seq clusters from (**b**). **h** Violin plots of gene expression module scores for each cell from scRNA-seq data of sgCD8A and sgKAT5KO GSC-0827 cells. oRG: outer radial glia. OPC oligodendrocyte precursor cells. Each data point = single cell. KS test was used to test significance ($p < 0.0001$)(1323 CD8 KO control cells; 840 KAT5 KO cells). Genes contained in each module are available in Supplementary Data 5. **i** Western blot validation studies of gene expression changes associated with loss of KAT5 activity. Protein extracts from GSC-0827 C13 cells were used from Dox+ or Dox- (7 days) conditions. Western blots were performed at least twice with similar results (except $n_{AURKA} = 1$, $n_{S100B} = 1$, $n_{KAT5} = 4$). Source data with exact $p$ values are provided with this paper as a Source Data file.

are common essential in cancer cell lines (depmap.org) and significantly overlap Myc bound genes in mouse ESCs[28] and MYC and E2F-transcription factor network targets (Fig. 5b, c), many of which have critical roles in cell cycle entry, S-phase, and mitosis. KAT5 target genes also include many transcription factors essential for GSC-0827 outgrowth which extend beyond common essential genes: *E2F3 (cell cycle entry)*[54], *MYC (cell cycle/growth)*[55], *NFE2L2* (redox homeostasis)[56], *POU3F2* (GBM stem-ness)[57], and *TCF7L2* (Wnt pathway/ G1/S transition)[58](Fig. 5c). KAT5 also targeted amplified genic in GSC-0827, such as *EGFR*, *MYC*, and *SEC61G* (Fig. 5c).

Analysis of cell state changes from scRNA-seq data in KAT5[on] and KAT5[off] tumor samples revealed that KAT5[off] cells displayed large increases in Neural G0 subpopulations, including a ~16-fold increase in cells in OPC-like cluster, representing 33% of the overall tumor cells, and a sixfold increase in the RG-like cells (Fig. 5d; Supplementary Fig. 12a). There were also concomitant, significant reductions in cell cycle and mesenchymal gene expression, both of which are associated with more aggressive tumor growth (Fig. 5e; Supplementary Fig. 12a).

To assess chromatin state changes, we used chromatin state discovery analysis (i.e., ChromHMM)[59], which integrates the chromatin marks and generate layered "emission states" across each region of the genome, identifying 8 emission states in GSC-0827 KAT5[on] tumors (Fig. 5f; Supplementary Data 13). These states incorporate activating H3K4me2 and H3K27ac marks (found at both promoters and enhancers[33]) and repressive H3K27me3 marks (required for long-term epigenetic stability of cell states during normal differentiation[34,35]) and encompass active, mixed valency, and repressive chromatin states. Of the 8 states, the two prominent active emission states, E3 (H3K4me2+H3K27ac) and E5 (H3K4me2 alone), shared 72% overlap in genic regions. These along with mixed valency sites had a higher probability of occurring in CpG islands and transcriptional start site of genes, in contrast to regions with only H3K27me3 sites (Fig. 5f).

Active E3 sites were found in 14,121 genes (Fig. 5f), which contained ~94% of KAT5 binding sites in C13 tumors and were, again, significantly enriched for MYC and also E2F target genes (Fig. 5g). These included classic E2F and MYC cell cycle target genes: *CCNA2*, *CCND1*, *CCNE2*, *CDC6*, *CDK4*, *E2F1-3*, *ODC1*, and *ORC1*. By contrast the "mixed valency" E4 state was associated with far fewer genes (2458) and only overlapped with ~11% of KAT5 bound genes (Fig. 5g). However, both E3 and E4 harbor essential KAT5-bound GSC-0827 transcription factors: *E2F3* (E3), *MYC* (E4), NFE2L2 (E3), *POU3F2* (E3/E4), and *TCF7L2* (E3). With regard to KAT5 targets, the results showed that KAT5 only binds a subset (~27%) of genes with activate chromatin marks, suggesting that KAT5/Tip60/NuA4 complex facilitates a "self-renewal" module in GSCs similar to its role mouse ESCs[28] rather than acting as a general transcription factor.

Further, in contrast to E3, the mixed valency E4 state had overlap with genic regions that become active in KAT5[off] cells, evidenced by transcriptional upregulated and/or gains of H3K27ac marks (~17% overlap) and/or losses of H3K27me3 marks (~42% overlap)(Fig. 5h). These included many genes associated with OPC and RG/Ac clusters in GSC-0827 tumors, including, for example, *OLIG1* and *OLIG2* (Fig. 5h) (Supplementary Data 14–15). Interestingly, H3K27me3 marks lost in KAT5[off] cells most often occurred within 1 kb of promoter region of affected genes (~47% of the time, compared to only ~10% for total peaks) (Supplementary Fig. 12c), suggesting a new and unknown regulatory mechanism (see Discussion).

Similar to in vitro results (Fig. 3c, d), there were multiple neurodevelopmental/GBM transcription factors associated with "activated" genes in KAT5[off] tumors (Fig. 5h; Supplementary Fig. 12d). In addition to HEY1, ID4, NPAS3, and SOX9 called out above, OLIG1/2 and SOX8 (OPC/oligodendrocyte specification)[60–63] were also observed. Moreover, regions gaining H3K27ac marks in KAT5[off] tumors showed enrichment for the degenerate Sox transcription motif (i.e., CWTTGT) (Supplementary Fig. 12e; Supplementary Data 16), which is required for Sox transcription factor binding, transcriptional activity, and neurodevelopmental cell patterning/function[64]. By contrast, H3K27ac peaks enriched in KAT5[on] tumors contained AP-1 complex motifs, which play key roles in mediating MAPK-pathway transcriptional responses[65] and mesenchymal and inflammatory gene expression in multiple cancers, including GBM[66,67] (e.g,. *CD44*) (Supplementary Fig. 12e). ATF4, essential to GSC-0827 cells (Fig. 5c, g), is part of this complex[65].

Callouts for multiple mixed valency regions affected by *KAT5* inhibition are shown in Fig. 5j and Supplementary Fig. 12f. One explanation of chromatin sites having both active and repressive marks is that they arise from different subpopulations within the bulk tumor sample, rather than being truly "bivalent". These populations, in turn, would have been derived from the same single stem cell clone (i.e., C13).

In addition, KAT5 activity affected predicted superenhancers, or regions containing clusters of enhancer (<12.5 kb apart) defined by H3K27ac marks[68] (Supplementary Fig. 12g-h; Supplementary Data 17). For KAT5[on] tumors, 521 superenhancer regions were identified that were associated with genes implicated in GBM mesenchymal cell states (e.g., *BCL3*, *CD44*, *RGS16*), GBM survival (*SEC61G*, *VGF*), neurogenesis/stemness (*PBX1*), and cell cycle (*CDK6*) (Supplementary Fig. 12g). These included 69 sites that were also bound by KAT5 (e.g., *CDK6*, *EGFR*, *SEC61G*). For KAT5[off] tumors, 218 superenhancer regions were associated with many of the OPC and RG/Ac genes highlighted above, including *BCAN*, *C1ORF61*, *GNG7*, *SMOC1*, and *SOX8*, as well others including angiotensinogen (*AGT*), a biomarker for bevacizumab response in GBM, *CSPG5*, a marker of OPC cells, *DBI*, associated with quiescent cancer cells, and *GFAP*, a key astrocytic marker found in the RG/Ac cluster (Supplementary Fig. 12h).

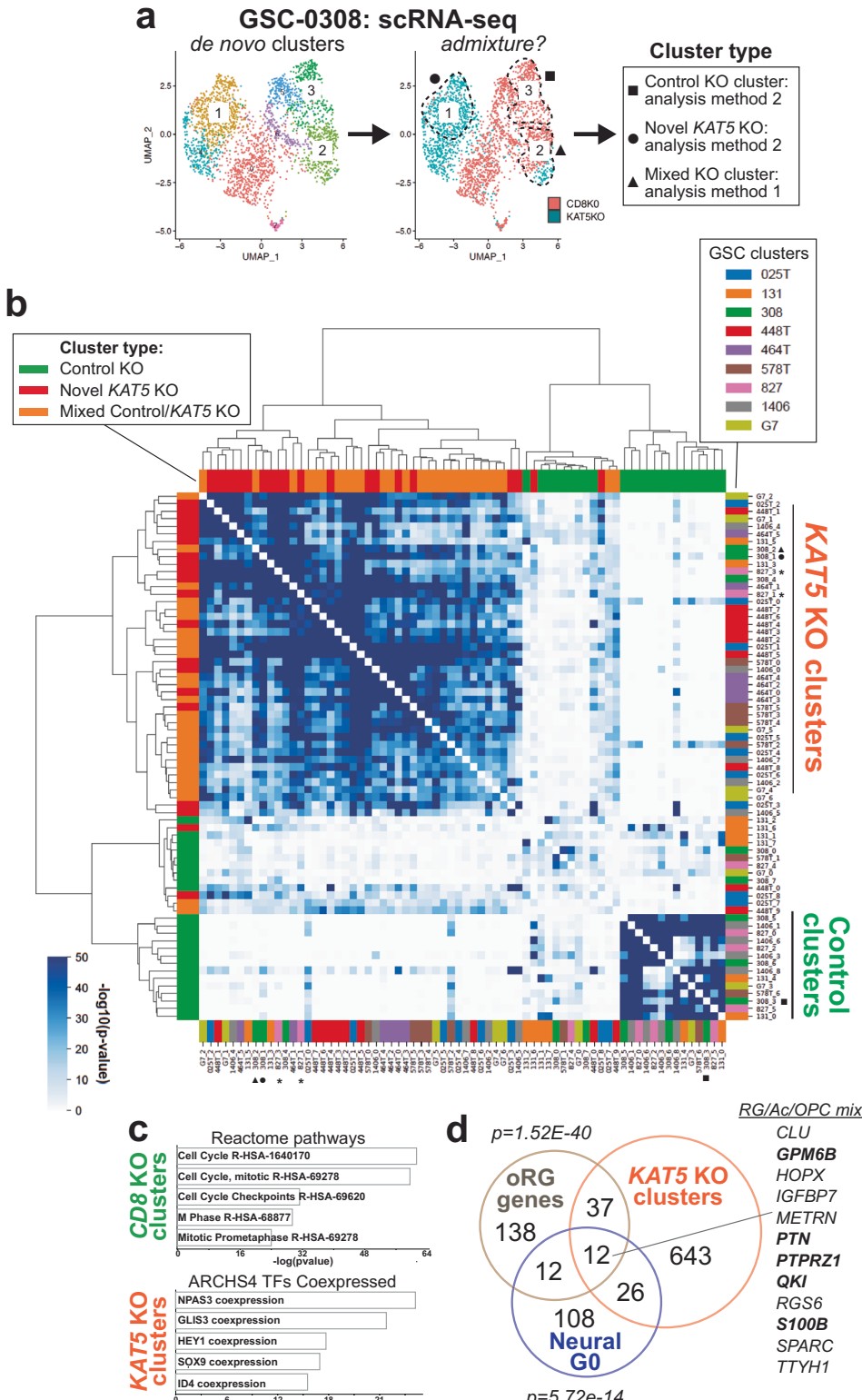

**Effects of attenuating KAT5 activity in GSC-derived tumors and on GSC invasiveness**

The above results demonstrate that KAT5 activity helps orchestrates E2F- and MYC-networks and likely other self-renewal modules (e.g., POU3F2, CD44) in GSC-derived tumors while suppressing the emergence of more indolent GBM tumor subpopulations. Consistent with this notion, C13 tumors deprived of Dox show significant induction in p27 and loss of Edu⁺ cells (Fig. 6a–d). Titrating Dox (0, 20, 200 and 2000 μg/ml) also resulted in significant reductions of tumor growth (Fig. 6e) and EdU incorporation (Fig. 6f). Moreover, attenuating KAT5 expression followed by standard of care (temozolomide + fractionated radiation) significantly reduced tumor growth post-SOC and provided a significant survival benefit (Fig. 6g, h; Supplementary Fig. 13a).

**Fig. 3 | KAT5-dependent cell state changes across 9 different GBM isolates.**
Single cell gene expression was used to assess changes in cell states in response cells after KAT5 inhibition in vitro. GSCs were collected for analysis post nucleofection of sgCD8 (control) and sgKAT5 RNPs after 5 days of growth in NSC self-renewal media. **a** Scheme for comparing *KAT5* versus *CD8* KO populations in scRNA-seq data. For each isolate, we co-embedded scRNA-seq data from *KAT5* KO and control *CD8* KO cells into a single merged scRNA-seq object. Then, de novo clustering was applied to each GSC line, and each cluster was examined for admixtures of *KAT5* and *CD8* KO cells. For assessment of gene expression changes, for clusters mainly composed of control *CD8* KO cells (*CD8* KO:*KAT5* KO cell ratios of ≥3) or *KAT5* KO cells (*CD8* KO: *KAT5* KO cell ratios of ≤.33), cluster marker gene enrichment (or "method 2") was used. For clusters composed of mixed populations of *CD8* and *KAT5* KO cells (cell ratios between 3 and. 33), DESeq2 was used to compare gene expression changes between *CD8* KO to *KAT5* KO cells within that cluster (or "method 1"). For both methods cutoffs were log2 fold change of 0.321 and adjusted pvalue for 0.05. The example in (**a**) highlights three clusters from each category. UMAPs for all 9 GSC isolates can be found in Supplementary Fig. 6. The full results from this analysis can be found in Supplementary Data S6. **b** Comparisons of *KAT5* KO dependent changes in cell states for 9 GSC isolate. The genes found significantly changed for each GSC cluster by either method 1 or 2 were compared using the Fisher's exact hypergeometric enrichment test and the -log10 *p* value and the overlapping marker genes were recorded for each pairwise comparison. Call out clusters from a are shown as filled circle, square, and triangle. *denotes novel *KAT5* KO clusters from Fig. 2. **c** Gene set enrichment analysis for gene enriched in CD8 KO. Comparisons of 272 genes enriched in CD8 KO clusters with S, S/G2, and G2/M phase genes form the ccAF cell cycle classifier. **d** Comparisons of 718 gene enriched in KAT5 KO clusters with Neural G0 genes and outer radial glial genes (from human fetal brain development). Genes were required to overlap with ≥50% of clusters to be considered for (**c**, **d**).

Because GBM tumors are highly invasive tumors and cell migration has been associated with quiescent GBM cells[69] (i.e., the "Go or Grow" hypothesis), we also examined whether inhibiting KAT5 activity affects GSC invasiveness. In 5 out of 6 GSC isolates examined, *KAT5* inhibition significantly decreased invasive behavior, while the other showed no significant difference (Fig. 6i, j; Supplementary Fig. 13b).

We also examined whether genes upregulated after KAT5 loss (Neural G0 genes) differ significantly in expression in clinical glioma tumor surgical samples by IDH1/2 mutation status and/or grade. We observed significantly higher expression in grade 2 and 3 IDH1/2 wt gliomas compared to grade 4 IDH1/2 wt tumors, and also higher expression in IDH1/2 mutant grade 2 and 3 gliomas compared to IDH1/2 wt grade 2 and 3 tumors (Supplementary Fig. 14). In addition, there is also significantly higher expression in grade 2 versus grade 3 tumors for astrocytomas, oligoastrocytomas, and oligodendrogliomas; and higher expression in each of these LGG tumor types when compared to GBM regardless of their grade (Supplementary Fig. 14). Neural G0 genes also significantly predicted survival in IDH1/2 mutant gliomas (*p* val. = 0.006) and show a strong tendency for IDHwt gliomas (*p* val. = 0.099). However, KAT5 expression on its own failed to predict survival in IDH1/2 mutant or wt gliomas.

### KAT5 activity in primary glioma cells
Finally, we wondered whether primary gliomas harbor populations with low KAT5 activity which would have molecular features associated with G0-like states. We first attempted to assay histone H4-Ac as a surrogate marker for KAT5 activity along with endogenous p27 and Ki67 levels by IHC in FFPE tumor samples and, also, by FACS analysis of freshly fixed and permeabilized samples (e.g., Fig. 1h). However, in each case, consistent co-detection proved problematic.

As an alternative, we assayed protein synthesis rate in freshly isolated tumor samples in combination with H4-Ac. As noted above, quiescent cells have lower protein synthesis rates compared to dividing cells[70] and also lower total cellular RNA content[71], which mainly consists of rRNA (~80%). With reduction of *KAT5*, we also observe significant reductions in rRNA levels, rRNA transcription via EU incorporation (which in mostly incorporated into rRNA), and protein synthesis rates (Fig. 7a; Supplementary Fig. 15a, b). Moreover, GSC p27[hi] cells have both low EU incorporation rates and lower H4-Ac levels, while loss of *KAT5* further reduces both (Fig. 1h; Supplementary Fig. 15b). In C13 cells, titration of Dox results in progressive loss of protein synthesis in line with H4-Ac loss (Fig. 7b), which is associated with higher percentages of p27+ cells (Fig. 1k). Interestingly, one principle likely at play here is conservation of cellular mass: as the protein translation rate falls, cells enter quiescence to conserve mass. For example, even after 14 days of KAT5 shut off, C13 cells maintain similar total protein content to KAT5[on] cells (Fig. 7c). By this logic, we expect primary tumor cells with lower synthesis rates to be more indolent and those with higher rates to be more aggressive.

For our assays, we collected 10 tumors (3 LGG IDH1/2mut, 5 HGG tumors, and 2 HGG IDH1/2 mut) and performed KAT5 activity and protein synthesis assays (30 min treatment with O-propargyl-puromycin (OPP)) in match cohorts of tumor cells (Fig. 7d). We used forward and side scatter, CD45 +, and viability stains to distinguish tumor cells from brain and immune cells.

These studies revealed that, first, KAT5 activity is dynamic in gliomas: KAT5[hi] and KAT5[low] populations are available in each tumor (Fig. 7e). Second, in general, LGG tumors have lower H4-Ac levels and protein translation rates than HGG gliomas (Fig. 7f–j; Supplementary Fig. 15c). Third, when LGGs recur as HGGs KAT5 activity and protein translation rates are concomitantly up regulated (Fig. 7f–i; Supplementary Fig. 15c). Finally, regardless of grade of tumor, KAT5 activity significantly correlates with protein translation rates in tumor cells, such that low KAT5 correspond to lower protein translation rates and vice versa (Fig. 7f–i).

We also examined cultured hNSCs and GSCs. Of note, *MYC* expression was sufficient to induce both KAT5 activity and protein translation in hNSCs (Fig. 8a), which mirrors increases in expression of *MYC* and *MYCN* and H4-Ac and AHA incorporation in GSCs compared to NSCs (Fig. 8a; Supplementary Fig. 15d–g). However, translation rates were also stimulated by expressing activated EGFR and AKT in the absence of a large increases in KAT5 activity (Fig. 8a). This is consistent with known roles for PI3K signaling in upregulating translation in cancer cells[72]. By contrast, affecting Rb or p53 function via expression of CyclinD1 + CDK4[R24C] and dominant-negative p53[73] did not alter KAT5 activity or translation rates in NSCs (Fig. 8a). Further, we evaluated sensitivity to the translation inhibitor homoharringtonine (HHT)[74], which is FDA approved for refractory CML[75] (Fig. 8b). NSCs with oncogene-induced high translation rates similar to GSCs showed significantly heightened sensitivity when compared to wt NSCs, similar to GSCs. NSC + *MYC* cells, in particular, displayed exquisite sensitivity. Taken together, the results show that KAT5 activity correlates with protein synthesis rate in tumors, GSCs, and NSCs, that HGGs have higher base line KAT5 activity and protein synthesis rates, and that HHT treatment may provide a therapeutic window for HGG tumors with high translation rates.

## Discussion
Here, we identified KAT5 as a key regulator G0 egress/ingress in GSCs. Our studies revealed that KAT5 activity promotes GSC self-renewal through coordinately regulating E2F- and MYC- transcriptional networks with ribosome assembly and protein translation. Inhibiting KAT5 causes simultaneous loss of transcription of key E2F- and MYC-targets while attenuating protein synthesis. As this occurs, GSCs and GSC-derived tumor cells transition into G0-like states with neurodevelopmental features commonly observed in OPC and RG cells, subpopulations of GBM tumors, and the computationally defined Neural G0 state (e.g., PTN/PTPRZ1)[2]. Moreover, titrating KAT5 activity in GSCs

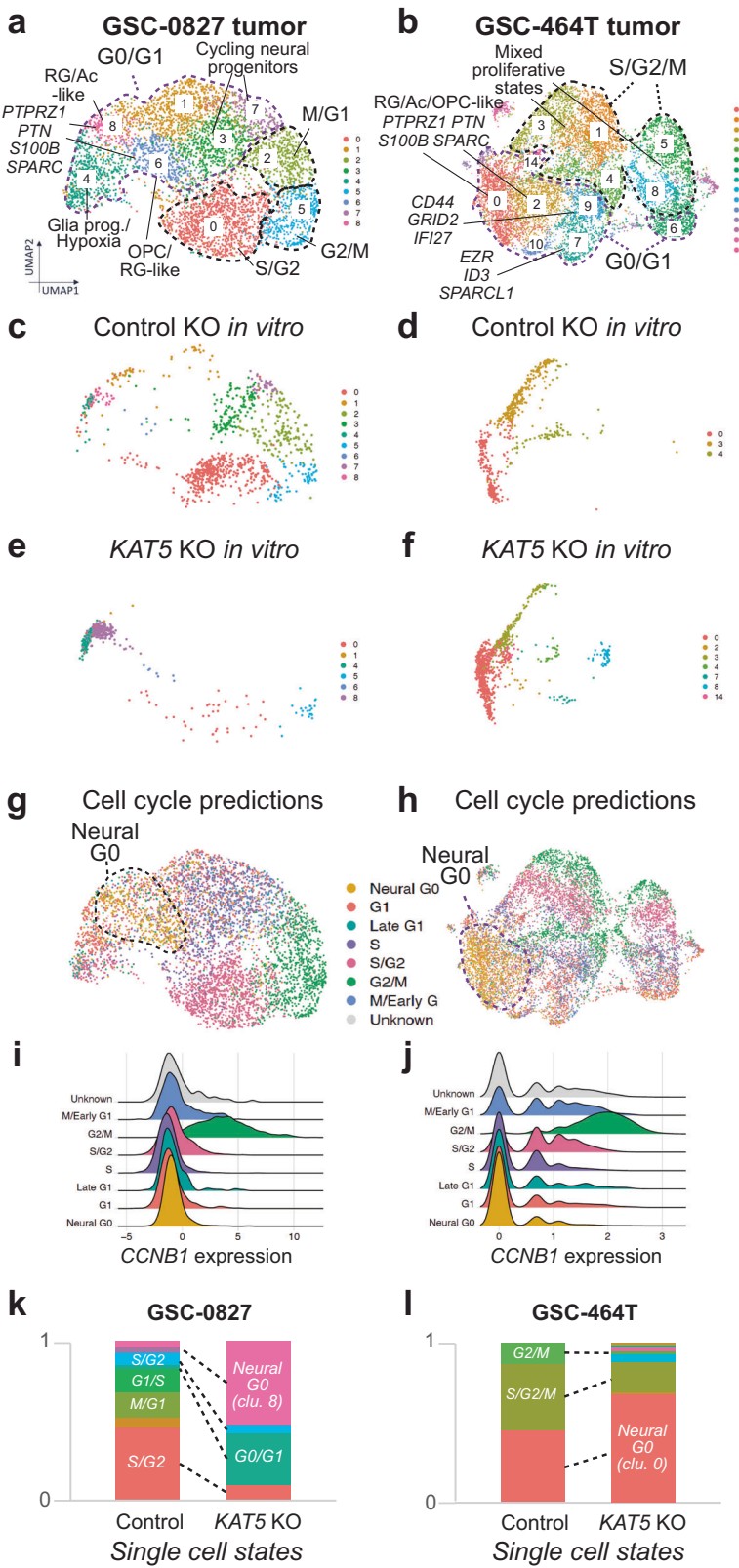

concomitantly attenuates G0 egress, protein synthesis, and tumor growth, revealing a dynamic interplay between cell cycle, growth regulation, and KAT5 activity (Fig. 8c).

This dynamic interplay could be observed in primary gliomas, where tumors have dynamic KAT5 activity, which directly correlates with protein synthesis rates. KAT5 activity and protein translation rate also differed by grade, where HGG were higher for both compared to LGGs (Fig. 7). In vitro, we found that *MYC* expression in NSCs was sufficient to induce KAT5 activity and translation rates as well as exquisite sensitivity to the translation inhibitor homoharringtonine (Fig. 8a). Previous work has established roles for PI3K signaling in upregulating translation rates via AKT/mTor dependent regulation of 4EBP1[72]. We observe a similar phenomenon in NSCs with EGFR + AKT expression, where

**Fig. 4 | *KAT5* KO triggers gene expression changes in GSCs consistent with predicted quiescent states in patient-derived xenograft tumors. a, b** Projections of scRNA-seq data for GSC-0827 and GSC-464T tumor references, respectively. Data was visualized using uniform manifold approximation and projection (UMAP) for dimensional reduction of data and generation of de novo cell-based clusters (Becht et al. 2018). Overview of experiment, filter cutoffs, and QC analysis are available in Supplementary Fig. 7. Supporting data includes: top enriched genes for each cluster; gene expression modules; and individual and gene set expression profiles (Supplementary Data 7-10; Supplementary Figs. 8–11). **c–f** Cells from scRNA-seq analysis performed on GSC-0827 and GSC-464T *CD8* and *KAT5* KO cells. The tumor references from (**a, b**) were used for mapping cells from scRNA-seq analysis performed on GSC-0827 and GSC-464T CD8 and KAT5 KO cells passed quality control metrics. **g, h** Cell cycle predictions using ccAFv2 computational classifier for (**a, b**), respectively. **i, j** Cyclin B1 gene expression for each predicted phase of the cell cycle from (**g, h**), respectively. **k, l** Relative proportions of mapped single cells appearing in clusters from (**c–f**) for GSC−0827 and GSC-464T cells, respectively.

translation was stimulated in the absence of upregulation KAT5 activity (Fig. 8a).

Both MYC and the KAT5/Tip60 complex regulate RNA polymerase I-dependent transcription of rDNA repeats, which, in turn, is rate limiting for ribosome assembly and protein translation[76–80],. We confirm that KAT5 controls rRNA transcription and also show that p27[hi] G0 GSCs have lower rRNA transcription rates than other cell cycle phases. Thus, it seems likely that MYC and KAT5/Tip60 have causal roles in regulating protein synthesis and promoting HHT sensitivity in HGG tumor cells. By contrast, LGGs have lower KAT5 activity than HGGs and lower expression of cell cycle genes (Supplementary Fig. 14).

Is KAT5/Tip60 itself a reasonable therapeutic target? Multiple small molecule KAT5 acetylase inhibitors have been developed that show promise in sensitizing cancer cells to chemotherapeutics or radiation (rev in ref. 81). In our hands, current KAT5 inhibitors trigger non-specific toxicity in GSCs at effective doses observed for other cell types. However, it seems likely there is therapeutic window for KAT5 inhibition given its increased activity in high grade tumors and also likely addiction to protein synthesis in KAT5[hi] tumor cells (Fig. 7). In addition, preclinical models demonstrated significant beneficial effect between KAT5 removal and SOC therapy (Fig. 6h). However, future experiments will need to address which KAT5/NuA4 complex activities are required for our observed phenotypes. In addition to its protein/histone acetylase activity, the EP400 subunit of the complex catalyzes ATP-dependent incorporation of histone H2A.Z into chromatin, which can affect gene transcription separately from its histone acetylase activity[82,83]. Further, KAT5 is highly expressed in the cerebellum (Supplementary Fig. 16) and has non-MYC/E2F functions which may represent on-target liabilities. Thus, one possibility is that KAT5 activity (i.e., H4-Ac staining) could be used as a biomarker for sensitivity to HHT (or a biosimilar). Of note, HHT has been shown to cross the BBB in phase I clinical trials[84].

One reason GBM tumor cells have heightened sensitivity to HHT may be due to its effect on the activity of short-lived proteins, including MYC and other key cell cycle (e.g., cyclins). However, HHT has also been shown to bind directly to NF-κB repressing factor (NKRF), which can affect MYC promoter activity[85]. Future work will be required to investigate these and other possibilities.

The KAT5/NuA4 functions as a transcriptional co-activator, whose activities are coordinated with multiple transcription factors, including, AR/ER[26,86], E2F proteins[27], and MYC[28,87]. In mouse ESCs, KAT5 co-regulates an ESC-specific c-Myc network of genes critical for their self-renewal, with minimal overlap with other pluripotency factor networks (Nanog, Oct4, and Sox2)[28]. Our results are consistent with this result. In GSCs, KAT5 activity acts as a regulatory module for subset of genes of GSC-0827 tumors with activated chromatin marks—including E2F and MYC transcriptional networks as well as AP-1 (Activator Protein-1) transcription factor targets, which play key roles in promoting cell cycle, Myc expression, inflammation, and mesenchymal gene expression in multiple cancers, including GBM[66,67,88]. In transformed mouse NSCs, loss of Fosl1 activity (an AP-1 subunit) was shown to cause loss of expression of mesenchymal genes and upregulation of neurodevelopmental genes (e.g., OPC, RG/Ac)[67]. This supports the notion that KAT5 activity is coordinated with AP-1 activity. We find that KAT5 binds to promoters of specific transcription factors required for GSC outgrowth: *ATF4, NEF2L2, POU3F2, TCFL2, YY1*, etc. (Fig. 5c), suggesting broad rewiring effects of KAT5 activity on the chromatin and transcriptional landscape. Future work will need to determine the how these transcriptional networks and changes in chromatin topography are integrated with GSC self-renewal and quiescence programs.

Of note, neither *MYC* knockdown or RNA Pol I inhibition (with CX-5641) fully recapitulated KAT5[off] phenotypes, only causing modest p27 induction (e.g., <1.6-fold compared >2.5-fold for KAT5 loss). *MYC* inhibition in GBM cells has been previously shown to cause apoptosis and trigger mitotic catastrophe, rather than G0-arrest[89]. Similarly, targeting ribosome subunits or inhibiting translation caused significant cell death rather than mainly G0 arrest. Perhaps down regulation of KAT5 causes a more physiological attenuation of self-renewal networks along with protein synthesis both increases the likelihood of G0 ingress.

One lingering question is how loss of *KAT5* triggers could increase the expression of genes associated with neurodevelopmental states? Because KAT5 regulates MYC expression, one possibility is that loss of KAT5 rewires the network of MYC-associated proteins that heterodimerize with MAX, MYC's obligate dimerization partner for E box binding and stimulation of transcription at target genes[90]. In other developmental contexts transitions from proliferation to differentiation are accompanied by changes in composition of MAX heterodimers, where MYC is switched out for alternate MAX partners such as MXD1 to trigger gene repression and differentiation[91–93].

An alternative hypothesis is that key regulators of neurodevelopmental states simply have longer half-lives in G0 states (e.g., similar to p27). In this scenario, these factors would tend to out compete other transcriptional programs in G0, giving rise to the observed transcriptional patterns. Future work will need to distinguish these possibilities.

Additionally, KAT5 was recently implicated is promoting GBM tumor growth by upregulating cholesterol synthesis (via acetylation of HMGCS1)[94], lipolysis of lipid droplets (via acetylation of choline kinase)[95], and EGFR-PI3K activity (via PFKP acetylation)[96]. These results further underscore the notion that high KAT5 activity in GBM cells can affect a diverse array of cell functions that drive aggressiveness.

In summary, we have provided here a framework for studying quiescent/G0-like states in GBM and identified mechanisms by which these quiescent/G0-like states are maintained by the histone acetylase KAT5. We provide evidence that G0-like states are relevant in human brain tumors and identify potential strategies for targeting KAT5[hi] tumor populations.

## Methods
### Key reagents and resources are available in Supplementary Data 19

**Ethical statement.** This research was approved by the Fred Hutch Institutional Animal Care and Use Committee (Protocol # 202100013) and complies with all required ethical regulations.

**Cell culture.** Patient tumor-derived GSCs were provided by Drs. Jeongwu Lee (Cleveland Clinic), Do-Hyun Nam (Samsung Medical Center, Seoul, Korea), and Steven M. Pollard (University of Edinburgh). Isolates were cultured in NeuroCult NS-A basal medium (StemCell Technologies) supplemented with B27 (Thermo Fisher Scientific), N2

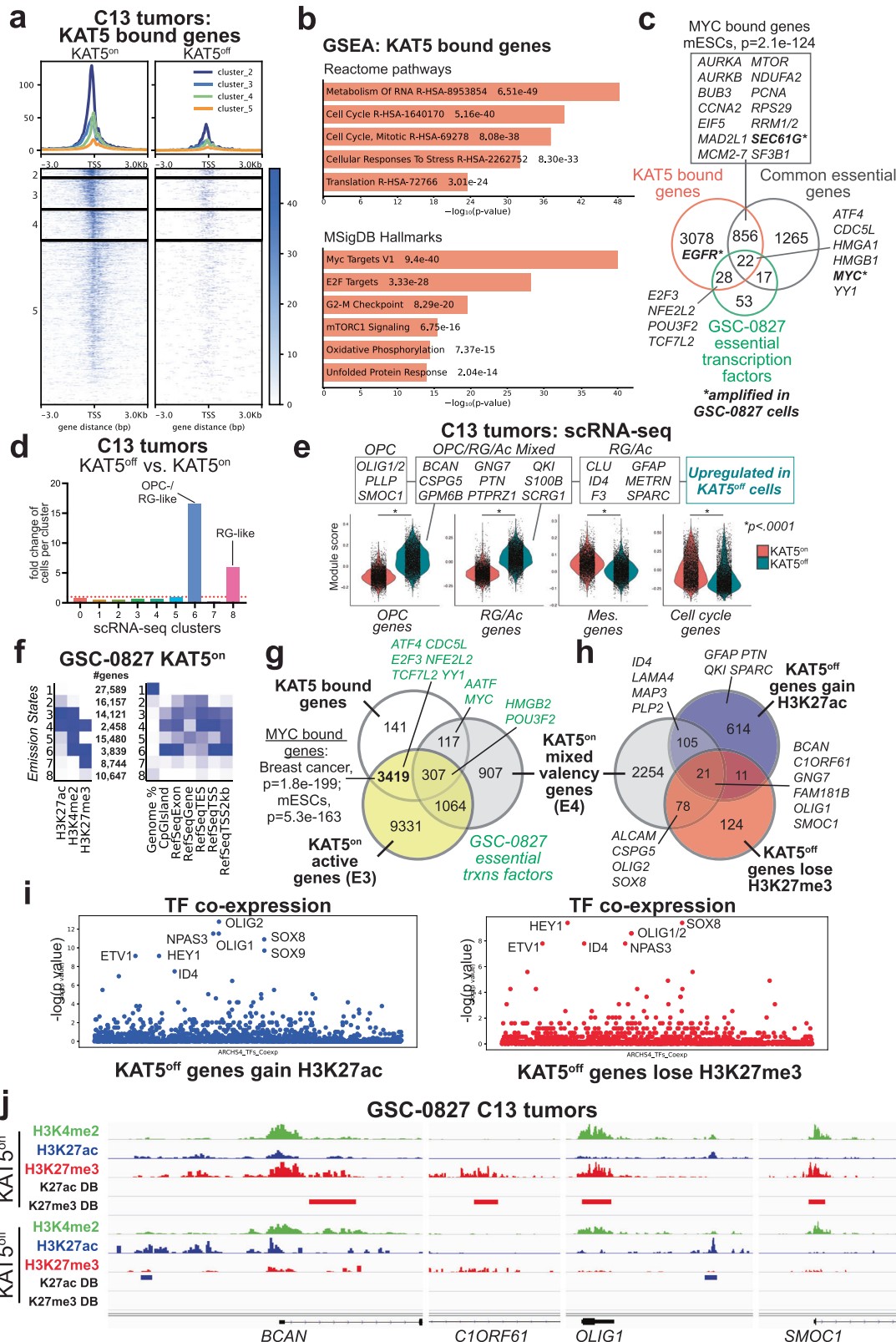

(homemade 2x stock in Advanced DMEM/F-12 (Thermo Fisher Scientific)), EGF and FGF-2 (20 ng/ml) (PeproTech), glutamax (Thermo Fisher Scientific), and antibiotic-antimycotic (Thermo Fisher Scientific). Cells were cultured on laminin (Trevigen or in-house-purified)-coated polystyrene plates and passaged as previously described[12], using Accutase (EMD Millipore) to detach cells.

**GSC tumors.** NSG mice (Jackson Labs #005557) used in this study were kept in a 12 h light/dark cycle at 72°F and 40-60% relative humidity with food and water ad libitum, in the Fred Hutchinson Cancer Center (FHCC) vivarium. All animal experimental procedures were performed with approval of the FHCC Institutional Animal Care and Use Committee (Protocol # 202100013). All procedures followed guidelines

**Fig. 5 | Analysis of KAT5 target genes, gene expression changes, and epigenetic patterning associated with KAT5$^{on}$ and KAT5$^{off}$ states in GSC-0827 tumors.** **a** KAT5 binding associated with transcription start sites in GSC-0827 tumors. An antibody recognizing the V5 epitope tag was used to perform CUT&TAG on KAT5-V5 ectopically expressed in C13 Dox + /KAT5$^{on}$ tumors. Dox-/KAT5$^{off}$ tumors (6 days Dox0-) were used as a control. K-mean clustering for KAT5$^{on}$ was used to define separate cluster classes shown (Supplementary Data 11). **b** Gene set enrichment analysis for KAT5 bound genes from clusters 2, 3, and 4 from (**a**). **c** Comparisons of KAT5 bound genes with common essential genes (depmap.org) and essential transcription factors in GSC-0827s. **d** Fold change of cells in each scRNA-seq cluster for KAT5$^{off}$ tumors. **e** Violin plots of gene expression module scores for each cell from scRNA-seq data of Dox+ vs. Dox- GSC-0827 C13 tumors. Each data point = single cell. KS test was used to test significance ($p < 0.0001$)(4951 Dox+ cells; 5507 Dox- cells). Genes associated with each model are available in Supplementary Fig. 12 and Supplementary Data 5. **f** ChromHMM analysis of genomic regions in KAT5$^{on}$ C13 tumor cells showing 8 possible chromatin states (i.e., emission states) for H3K4me2, H3K27ac, and H3K27me3, the associated number of genes, and emission state region genomic annotations. Genome% = intergenic space; TES = transcription end sites; TSS = transcription start sites. The darker blue color corresponds to a greater probability of observing the mark in the state. The full data set is available in Supplementary Data 12. **g, h** Overlap of genes associated with emission state E4 from (**f**), which display both activating and repressive chromatin marks, and those with significant changes H3K27ac and H3K27me3 after loss of KAT5 activity in GSC-0827 tumors. DiffBind and DESeq2 were used to score significant changes in chromatin marks (Supplementary Data 13,14, 15) from KAT5$^{on}$ and KAT5$^{off}$ C13 tumor samples ($n = 2$). Note: DiffBind could not be performed on H3K4me2 marks for C13 tumor samples because one replicate failed to produce sufficient quality data. **i** Enrichment for genes associated with expression of human transcription factors (TFs). Left panel: genes enriched for H3K27ac regions after loss of KAT5 activity. Right panel: genes depleted for H3K27me3 regions after loss of KAT5 activity. The analysis was performed using the Enrichr pipeline (Kelshov et al., 2016) using the hypergeometric test by comparing the top 300 genes associated with specific human transcription factors in the ARCHS4 database (Lachmann, et al. 2018). **j** Examples of multivalent genes in KAT5$^{on}$ C13 tumors that upon Dox withdrawal gain H3K27ac and lose H3K27me3 marks (i.e., overlapping genes from **h**). Additional examples including GNG7, SLIT1, and SOX8 are shown in Supplementary Fig. 12.

outlined in the National Research Council Guide for the Care and Use of Laboratory Animals. 100,000 GSCs were orthotopically xenografted into a single frontal cerebral hemisphere. GSCs were injected using stereo-tactic coordinates: 2 mm lateral from Bregma and 3.5 mm depth and grown for 3-12 weeks according to our previously published protocols[73,97,98]. We relied on MRI and clinical/neurological symptoms to evaluate the tumor burden. Once the tumor was visible by MRI, mice were monitored 3 times a week and then daily if neurological symptoms were observed. Animals that exhibited the following criteria for maximal tumor burden were euthanized and considered at endpoint: impaired mobility, inability to reach food and water, hunched posture, labored breathing and/or cyanosis (bluish ears, feet or mucous membranes), abnormal response to stimuli (slow to move/does not move or reacts excessively), body conditioning, and skin ulceration and/or necrosis, in the event of extracranial tumor growth. Maximal tumor burden was not exceeded in these studies. *Doxycycline dosage of mice:* 827-C13 cells were transplanted into NSG mice that were pre-dosed with Dox (2 mg/ml) into the drinking water (supplemented with 5% sucrose) 24 h in advance. Tumors were allowed to form in the continuous presence of Dox until they were detected by MRI, followed by enrollment into experimental cohorts (i.e., Dox + , Dox-, Dox + /SOC, Dox-/SOC). *EdU pulsing of mice:* 6–20 h prior to tumor harvesting mice were intra-peritoneally injected with EdU (100 mg/Kg). *MRI:* volume of interest was manually contoured using T2-weighted brain MRI scans (Bruker 1 T and 7 T scanner) for all animals using Horos Dicom Viewer software (horosproject.org). SOC treatments were started when tumors reached 2 mm^3 volume as analyzed by MRI, and 4 days after Dox withdrawal in Dox-/SOC cohorts. SOC treatments consisted of 5 consecutive days of 50 mg/Kg TMZ IP injections concomitantly with fractionated 2 Gy/day x-ray irradiation on days 1 and 3 (4 Gy total). Mouse heads (the rest of the mouse bodies were lead shielded) were irradiated while under Isoflurane anesthesia using Precision X-RAD 320 Biological Irradiator (https://precisionxray.com/).

**p27 and FUCCI reporters.** The p27 reporter was constructed after (Oki et al., 2014), using a p27 allele that harbors two amino acid substitutions (F62A and F64A) that block binding to Cyclin/CDK complexes but do not interfere with its cell cycle-dependent proteolysis. This p27K⁻allele was fused to mVenus to create p27K⁻-mVenus. To this end, the p27 allele and mVenus were synthesized as gBlocks (IDT) and cloned via Gibson assembly (NEB) into a modified pGIPz lentiviral expression vector (Open Biosystems). Lentivirally transduced cells were puromycin selected. P27-mVenus reporter cells were sorted for the presence of mVenus on an FACSAria II (BD) and normal growth was verified post-sorting. FUCCI constructs (RIKEN, gift from Dr. Atsushi

Miyawaki) were transduced into GSC-0827 cells and sorted sequentially for the presence of mCherry-CDT1(aa30-120) and S/G2/M mAG-Geminin(aa1-110) on an FACSAria II (BD). Normal growth was verified post-sorting and then the FUCCI GSCs were transduced with individual sgRNA-Cas9 and selected with 1 µg/mL puromycin. Cells were grown out for 21 days with splitting every 3-4 days and maintaining equivalent densities. Cells were counted (Nucleocounter NC-100; Eppendorf) and plated 3 days before analysis on an LSR II (BD).

**Lentiviral Production.** For virus production, lentiCRISPR v2 plasmids were transfected using polyethylenimine (Polysciences) into 293 T cells along with psPAX and pMD2.G packaging plasmids (Addgene) to produce lentivirus. For the whole-genome CRISPR-Cas9 libraries, 25 × 150 mm plates of 293 T cells were seeded at -15 million cells per plate. Fresh media was added 24 h later and viral supernatant harvested 24 and 48 h after that. For screening, virus was concentrated 1000x following ultracentrifugation at 6800 × g for 20 h. For validation, lentivirus was used unconcentrated at an MOI < 1.

**CRISPR-Cas9 screening.** For large-scale transduction, GSC cells were plated into T225 flasks at an appropriate density such that each replicate had 250-500-fold representation, using the Brunello CRISPR-Cas9 library[99,100] (Addgene) (as we have previously published[101]). GSCs were infected at MOI < 1 for all cell lines. Cells were infected for 48 h followed by selection with 2 µg/mL of puromycin for 3 days. Post-selection, a portion of cells were harvested as Day 0 time point. The remaining cells were then passaged in T225 flasks maintaining 250-500-fold representation and cultured for an additional 8 days. Genomic DNA was extracted using QiaAmp Blood Purification Mini or Midi kit (Qiagen). A two-step PCR procedure was performed to amplify sgRNA sequence. For the first PCR, DNA was extracted from the number of cells equivalent to 250-500-fold representation (screen-dependent) for each replicate and the entire sample was amplified for the guide region. For each sample, -100 separate PCR reactions (library and representation dependent) were performed with 1 µg genomic DNA in each reaction using Herculase II Fusion DNA Polymerase (Agilent) or Phusion High-Fidelity DNA Polymerase (Thermo Fisher). Afterwards, a set of second PCRs was performed to add on Illumina adaptors and to barcode samples, using 10–20 µl of the product from the first PCR. Primer sequences are in Supplementary Data 17. We used a primer set to include both a variable 1–6 bp sequence to increase library complexity and 6 bp Illumina barcodes for multiplexing of different biological samples. The whole amplification was carried out with 12 cycles for the first PCR and 18 cycles for the second PCR to maintain linear amplification. Resulting amplicons from the second

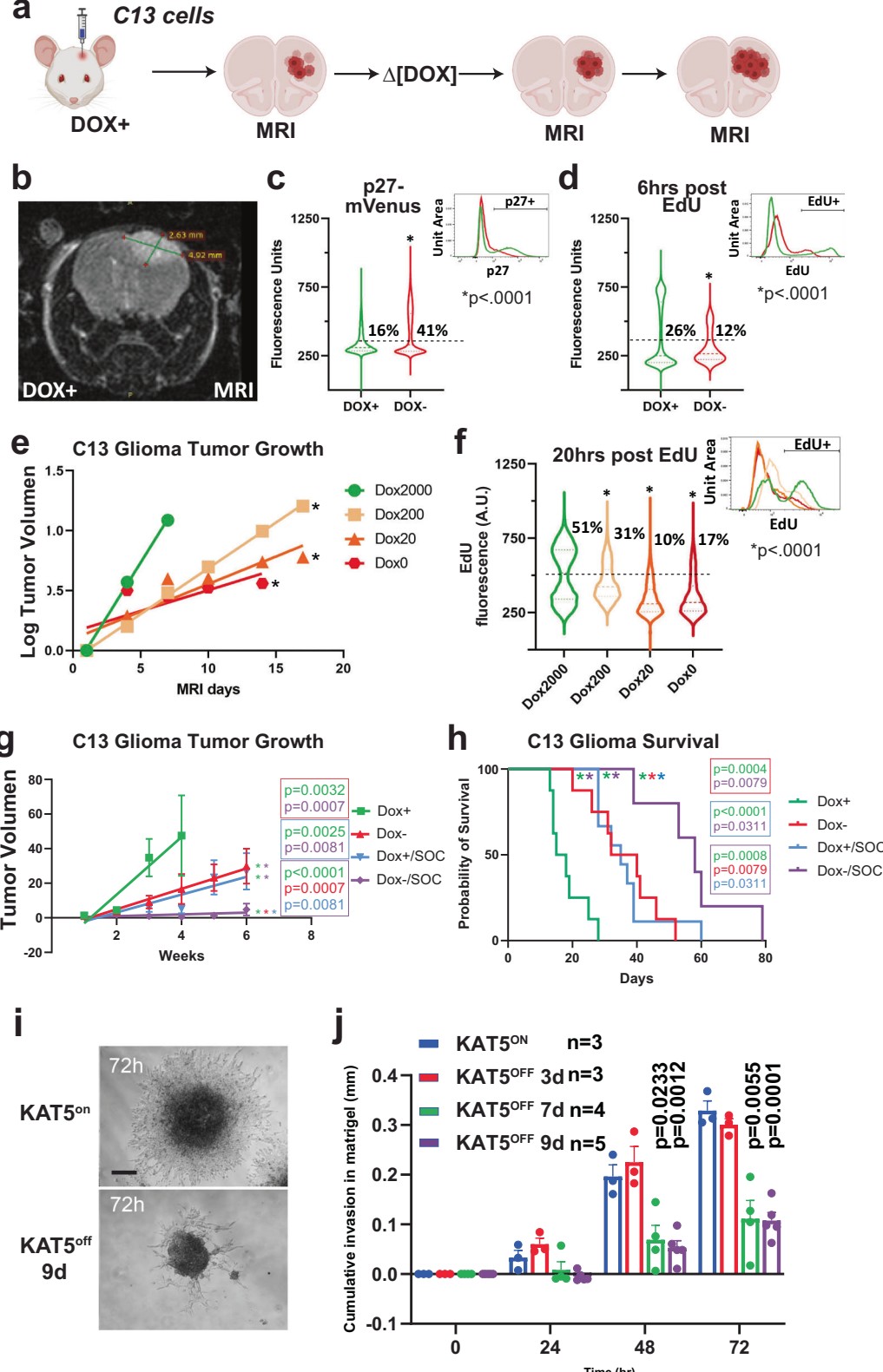

Nature Communications| (2025)16:4327                                                              13

PCR were column purified using Monarch PCR & DNA Cleanup Kit (New England Biolabs; NEB) to remove genomic DNA and first round PCR product. Purified products were quantified (Qubit 2.0 Fluorometer; Fisher), mixed, and sequenced using HiSeq 2500 (Illumina). Bowtie was used to align the sequenced reads to the guides[102]. The R/Bioconductor package edgeR was used to assess changes across various groups[103]. Similar to the whole genome CRISPR-Cas9 KO screen

above, we performed an outgrowth CRISPR-Cas9 screen for transcription factors that are essential for GSC growth. We used a transcription factor sgRNA library that contains 6813 sgRNAs targeting 1448 genes (4 sgRNAs per gene with a few exceptions) and 1000 non-targeting controls. Sequence-specific human transcription factors were curated in collaboration with James Park in Nitin Baliga lab at the Institute for Systems Biology, Seattle.

**Fig. 6 | Modulation of KAT5 activity in GSC-derived tumors and during GSC invasion assays. a** Scheme for using C13 GSC-0827 cells for creating PDX tumors in NSG mice. Created in BioRender. Paddison, P. (2025) https://BioRender.com/tehoi00. **b** Representative MRI image of mouse head with Dox-KAT5 tumor in cortex at 36 days post-injection. **c** Analysis of p27 levels after Dox withdrawal in C13 tumor containing mouse. KS test was used to test significance ($p < 0.0001$). **d** Analysis of EdU incorporation (6 h) after 7 days of Dox withdrawal in C13 tumor containing mouse. KS test was used to test significance ($p < 0.0001$). **e** Tumor growth as assessed by volume using MRI in C13-induced PDX tumors after switching drinking water to concentration of Dox indicated (µg/mL). Linear regression analysis was used to assess significance ($p$ val overall $= 0.001$; $p$ val2000 vs. 200 $< 0.0001$; $p$ val2000 vs. 20 $= 0.0049$; $p$ val2000 vs. 0 $= 0.0183$). **f** EdU incorporation for 20 h at 14 days after Dox concentration from (**e**). KS test was used to test significance ($p < 0.0001$). **g** C13 xenografts were treated with standard of care (SOC) in the presence (Dox + ) and/or absence (Dox−) of KAT5. Tumor growth was assessed by volume using MRI. The plot represents a linear analysis with the first MRI time point at enrollment of mice in experimental cohorts. The asterisk denote statistically significant differences between pairs of slopes for the indicated experimental conditions by color (nDox + = 8, nDox- = 8, nDox + /SOC = 9, nDox-/SOC = 5). Plots show mean ± SEMs. with simple linear regression curve comparisons for significance. **h** Kaplan-Meier survival probability plot for tumor bearing mice from (**g**). Survival probability differences between paired comparisons was assessed by log rank (Mantel-Cox) test. The asterisk denote statistically significant differences between pairs of comparisons for the indicated experimental conditions by color. Average survival days gained: Dox- = 17.75; SOC/Dox + = 17.97; and SOC/Dox- = 39.55. **i, j** KAT5^on vs. KAT5^off C13 invasion assays. C13 cells were grown as spheres for 3 days in either 1µg/ml Doxycycline for KAT5^on or no Doxycycline for KAT5^off conditions and transferred to Matrigel covered wells for the specified number of days (3 d, 7 d, or 9 d). Phase-contrast images were captured every 24 h for 72 h. The area covered by invading cells was measured using FIJI. **i** Representative phase contrast microscopy images of invasion assay. Spheroids were embedded into Matrigel. Scale bar is 200 µm. **j** Quantification of (**e**). $n_{KAT5on} = 3$, $n_{KAT5off\ 3d} = 3$, $n_{KAT5off\ 7d} = 4$, $n_{KAT5off\ 9d} = 5$. Mean ± SEM shown (significance was tested using repeated measurements 2-way ANOVA with Giesser-Greenhouse correction and Sidak multiple corrections test; $p \leq 0.01$). Additional isolates are assayed in Supplementary Fig. 13. Source data with exact $p$ values are provided with this paper as a Source Data file.

**Cas9:sgRNA RNP nucleofection.** Knockout of endogenous genes in GSCs was performed and analyzed as detailed in ref. 104. Lyophilized chemically synthesized sgRNA (Synthego) was reconstituted to 100 pmoles/µL in nuclease-free 1X TE Buffer (Tris-EDTA, pH 8.0) and was used directly for RNP complexing or diluted to 30 pmoles/µL in nuclease-free water immediately before use, depending on the particular dosing. Purified sNLS-SpCas9-sNLS (Aldevron) was diluted from 61 pmoles/µL to 10 pmoles/µL in PBS (pH 7.4) immediately before use. To prepare RNP complexes, reconstituted sgRNA was added to SG Cell Line Nucleofector Solution (Lonza), followed by addition of Cas9, to a final volume of 20 µL. A Cas9:sgRNA ratio of 1:2 was used, unless otherwise noted. Total dose of RNPs described in this paper refers to the amount of the limiting complex member (Cas9). The mixture was incubated at room temperature for 15 min to allow RNP complexes to form and then placed on ice until use. To nucleofect, $1.3$-$1.5 \times 10^5$ cells were harvested, washed with PBS, resuspended in 20 µL of RNPs, and electroporated using the Amaxa 96-well Shuttle System (Lonza) and program EN-138, similar to[105]. After nucleofection, cells were recovered in pre-warmed culture media and plated onto 12-well or 6-well plates. Media was changed 12–24 h after nucleofection.

**CRISPR editing analysis.** Nucleofected cells were harvested at designated timepoints and genomic DNA was extracted (MicroElute Genomic DNA Kit, Omega Bio-Tek). Genomic regions around CRISPR target sites were PCR amplified using primers located (whenever possible) at least 250 bp outside cut sites. After size verification by agarose gel electrophoresis, PCR products were column-purified (Monarch PCR & DNA Clean-up Kit, New England BioLabs) and submitted for Sanger sequencing (Genewiz) using unique sequencing primers. The resulting trace files for edited cells versus control cells (nucleofected with non-targeting Cas9:sgRNA) were analyzed for predicted indel composition using the ICE web tool[106].

**Flow cytometry.** GSC cells that incorporated EdU while alive (2–24 h, 10–2 µM) EU (1 µM for 1 h) or AHA (100 µM for 1 h after cells were grown in Methionine-free SILAC media for 30 min.) were fixed (4% PFA) and permeabilized (0.1% Saponin) and subjected to Click-iT Plus chemistry detection prior to analysis by flow cytometry. In some cases, the same cells were also stained for H4-panAc (1:100) for KAT5 function evaluation, DAPI (0.001 µg/ul) for DNA content analysis, or Pyronin Y for RNA content analysis. H4-panAc staining was done in the presence of 0.3% Triton X-100 and 5% normal goat serum for 1 hr at room temperature and a secondary antibody conjugated to an Alexa Fluor was use for primary antibody detection (for 30 min. at room temperature) during flow analysis. Processed cells were flow cytometry analyzed immediately using either a BD FACSymphony A5 or BD LSRFortessa X-50 machine. Results were analyzed using FlowJo software. For experiments using sorted p27-high populations from GSC-p27-mVenus reporter cells, a gate at ~20% p27 high was used to isolate these populations. Either a BD FACSymphony S6 or Sony MA900 cell sorters were used. Supplementary Figs. 17, 18 show FACS gating strategies for each assay.

**Histone H4 Acetylation and OPP Analysis of Primary Glioma Samples.** Over the course of 2 years, 10 patient glioma tumors were collected and assayed: 3 LGGs (UW33, UW36, UW44), 5 HGGs (UW27, UW31, UW34, UW38, UW40), and 2 HGGs IDH1/2 mutant (UW26, UW44), which represent recurrent LGGs (Supplementary Data 16). Freshly resected tumors were dissociated by a combination of mechanical and enzymatic methods using a brain tumor dissociation kit (Miltenyi # 130-095-942) according to the manufacture's protocol. Cells were counted, and while alive, they were incubated with OPP (2 µM final, 1 million cells in 1 ml NSC media, in a low binding tube at 37°C for 30 min). Then cells were washed in warm NSC, and slowly frozen (1°C cooling/1 min.) in NSC media supplemented with 10% DMSO. Cohorts of HGG^WT and LGG and/or HGG^MUT were processed together in order to use the HGG^WT as normalization among cohorts that were collected over 2 years. Frozen cells were thawed using a method that recovers a large percentage of viable cells after freeze/thaw (frozen cells were thawed at 37 °C for 2 min., then warm NSC media was added dropwise to cells by doubling the volume every minute to a total volume of 32 ml, starting from one 1 ml frozen cell vial). While cells were alive, a CD45 antibody was used to surface stain immune tumor populations, a viability fixable Zombie dye (BioLegend) was used to determine live cells, then fixed in 4% PFA, permeabilized, and processed for Click-iT chemistry to detect OPP, and intracellularly stained for H4Ac. Processed cells were flow cytometry analyzed immediately using either a BD FACSymphony A5 or BD LSRFortessa X-50 machine. FSC-A and SSC-A plots were used to gate the tumor populations and eliminate cell debris, FSC-A and FSC-H plots were used to gate singlet populations, a gate on Zombie dye negative cells was used to isolate live cells, followed by a CD45 negative gate to isolate the non-immune tumor cell populations that were entered into the downstream analysis to evaluate protein synthesis rates (OPP + ), and KAT5 activity (H4Ac+).

**Western blotting.** Cells were harvested, washed with PBS, and either immediately lysed or snap-frozen and stored at -80°C until lysis. Cells were lysed with RIPA buffer (Thermo Scientific cat# 89900), 1X complete protease inhibitor cocktail (complete Mini EDTA-free, Roche)

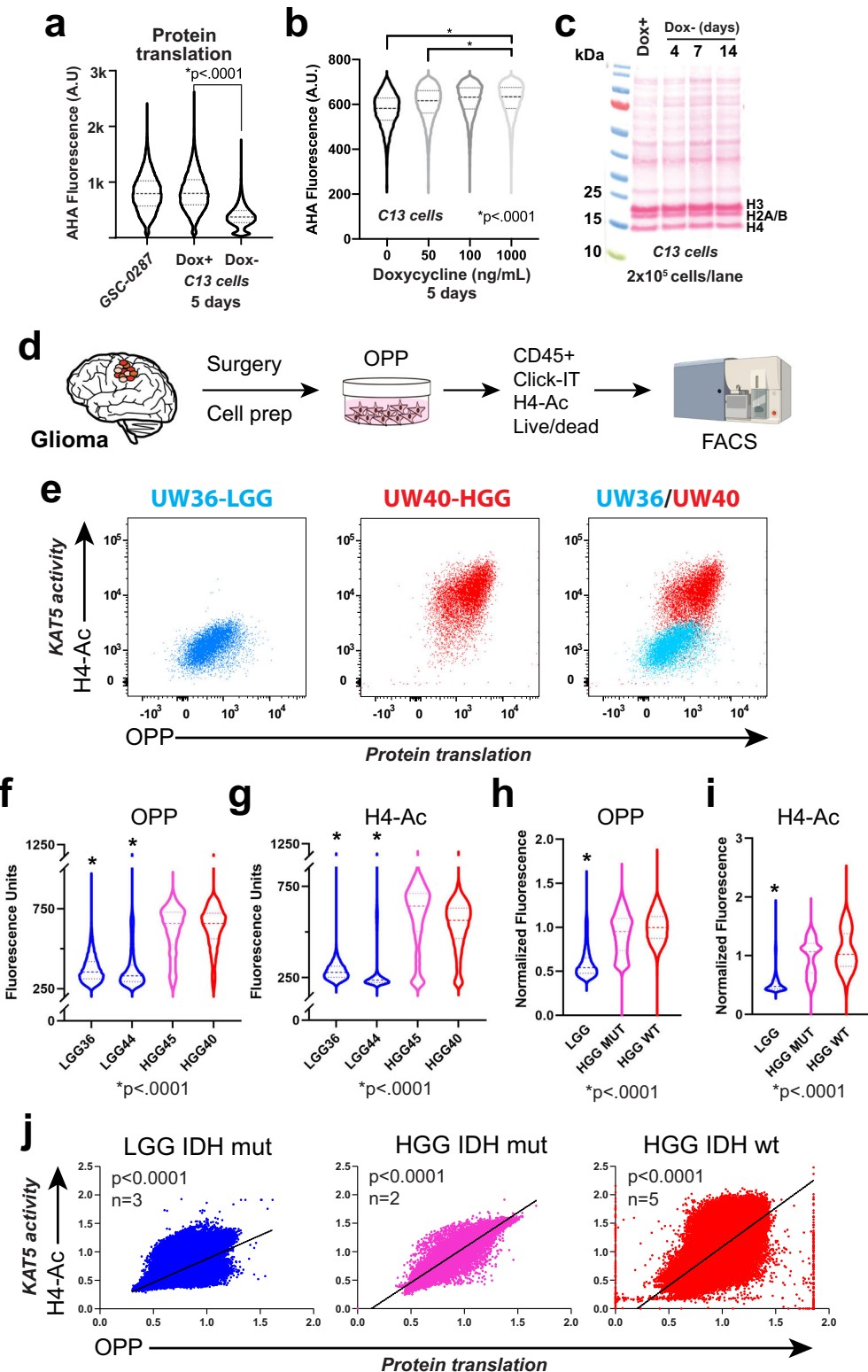

and 2.5U/µL benzonase nuclease (Novagen) in RIPA buffer supplemented with 1 mM final MgCl$_2$ concentration, at room temperature for 15 min. To enhance histone modification detection, a total histone extraction kit was used according to the manufacturer's protocol (Epigentek # OP-0006). Cell lysates were quantified using Pierce BCA protein assay reagent and proteins were loaded onto SDS-PAGE for western blot. The Trans-Blot Turbo transfer system (Bio-Rad) was used

according to the manufacturer's instructions. Ponceau S was used to visualize total proteins on the western blot membranes prior to the blocking step. An Odyssey infrared imaging system was used to visualize blots (LI-COR) following the manufacturer's instructions.

**Creation of Doxycycline controllable KAT5 GSC-0827 cells.** A KAT5-V5 tag ORF was cloned into the Tet-inducible expression

**Fig. 7 | Assessment of KAT5 activity and protein translation rates in primary glioma tumor samples. a** Examination of protein synthesis as measured by L-azidohomoalaine (AHA) incorporation in parental 827 cells and for KAT5 inhibited C13 cells after 7 days of Dox withdrawal. KS test was used to test significance. Note: AHA was used for these experiments because OPP (below) cannot be used in puromycin resistant cells. KS test was used to test significance. *$p$ val < 0.0001 ($n \geq 16970$). **b** Examination of protein synthesis via AHA incorporation in C13 cells grown in shown concentrations of Dox for 5 days. KS test was used to test significance. *$p$ val < 0.0001 ($n \geq 5703$). **c** Examination of total protein content in C13 cells using ponceau staining of total protein extract from 200,000 C13 cells Dox withdrawal of 0, 4, 7, and 14 days. Protein measurement was performed three times with similar results. **d** The scheme used for examining protein synthesis using O-propargyl-puromycin (OPP) incorporation in primary tumor cells followed by FACS-based assessment of OPP, histone H4 acetylation (i.e., KAT5 activity), CD45 +, and viability. Samples were dissociated, OPP-labeled, viably frozen, thawed, and

flow analyzed as a cohort. Tumor cells are CD45-. FACS machine cartoon created in BioRender. Paddison, P. (2025) https:// BioRender.com/o2c4y3l. **e** Plots of OPP versus pan-H4-Ac in LGG (UW36) and HGG (UW40). **f, g** Violin plots of OPP and pan-H4-Ac assay results, respectively, from two LGG (IDH1/2mut) (blue), one HGG (IDH1/2 mut) (pink) and one HGG (IDH1/2 wt) (red). Each flow event = single cell. KS test was used to assess significance ($p < 0.0001$)($n \geq 2751$); LGG vs either HGG WT or IDH1/2mut. **h, i** Violin plots of OPP and pan-H4-Ac assay results for 3 LGG (IDH1/2mut) (blue), two HGG (IDH1/2 mut) (pink) and 5 HGG (IDH1/2 wt) (red). FACS values are normalized in order for to compare across cohort groups. Each flow event = single cell. KS test was used to assess significance ($p < 0.0001$)($n \geq 59074$); LGG vs either HGG WT or IDH1/2mut.Supplementary Fig. 15 shows similar data for other individual tumors for LGG, HGG, and IDH1/2 mut HGG tumors. Supplementary Data 18 provides descriptions of each tumor sample used. **j** Combined of OPP versus pan-H4-Ac for LGG (Pearson r = 0.77), HGG (IDH1/2mut) tumors (Pearson r = 0.93), and HGG (Pearson r = 0.68), respectively; $p < 0.0001$ for each.

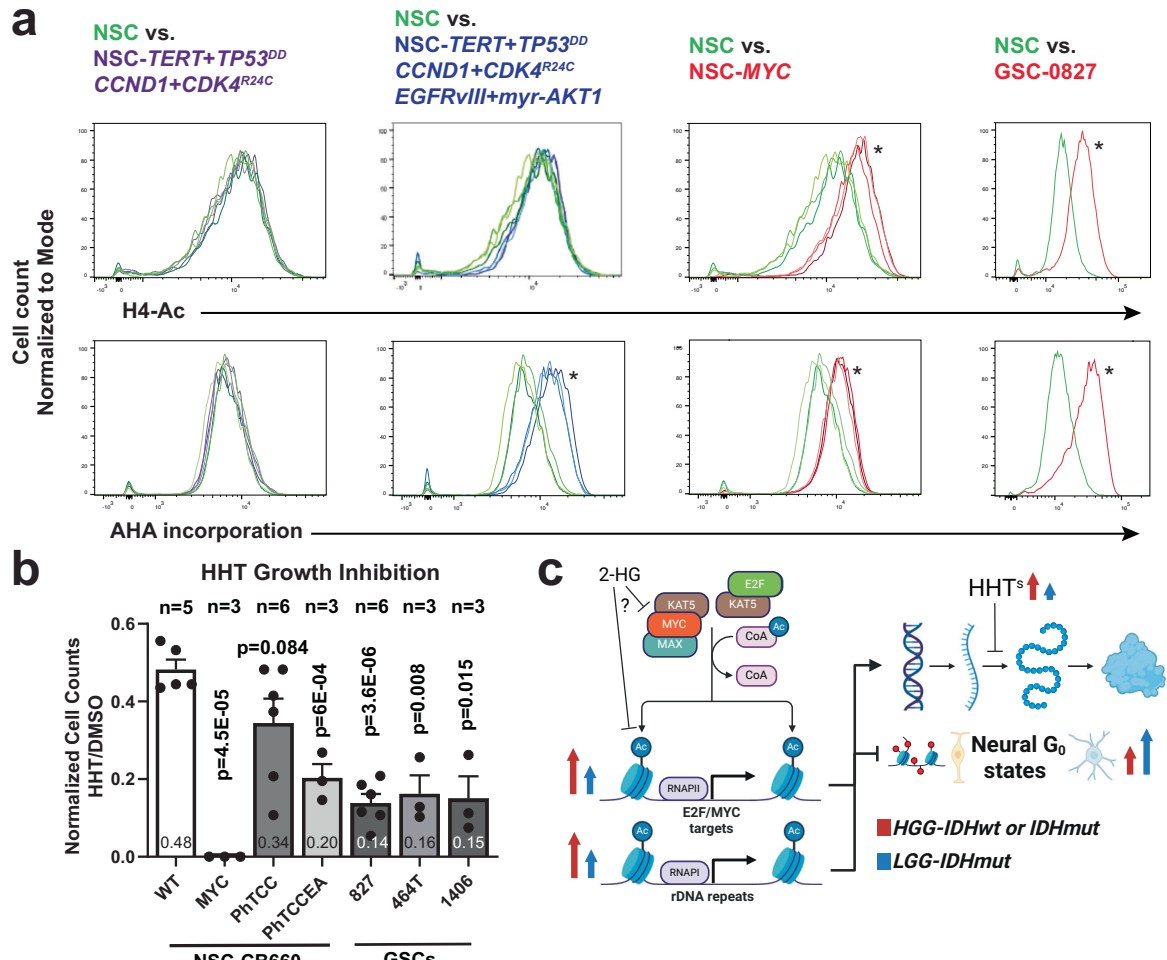

**Fig. 8 | Assessment of KAT5 activity, AHA incorporation, and homoharringtonine sensitivity. a** Flow-based assessment of H4-Ac and AHA incorporation in NSCs, transformed NSCs and GSC-0827 cells. **b** Homoharringtonine (HHT) sensitivity of NSCs, transformed NSCs, and GSCs. The approximate IC50 (100 nM) for NSC-CB660 cells was used for this assay. Cells were treated for 4 days 100 nM HHT (the -IC50 of NSC-CB660 cells) after which viability was measured ($n \geq 3$). Means and ± SEMs are shown, unpaired, 2 tailed t-tests were used for significance. Supplementary Fig. 16 shows MYC and MYCN expression as well as H4-Ac

and AHA incorporation for GSC-464T and GSC-1406. **c** Model of KAT5 function in HGG and LGG tumors arising from this work. LGGs have lower levels of KAT5 activity and cell cycle genes but appear to maintain similar levels of *MYC*, *MYCN*, and *KAT5* transcripts as HGGs (Supplementary Fig. 16). If true, one possibility is that the presence of excessive cellular 2-hydroxyglutarate, produced by oncogenic IDH1/2 mutant enzymes, keeps KAT5 activity. Created in BioRender. Paddison, P. (2025) https://BioRender.com/6xzam1p. Source data and exact *p* values are provided with this paper as a Source Data file.

retroviral plasmid pTight-TURB vector (MSCV-tetO7-mCherry-UBC-rtTA-Blast) via Gibson assembly a.GSC-0827/p27-mVenus reporter cells were infected with pTight-TURB-KAT5-V5 (3 rounds of infection over a 3 day period) and Blasticidin selected. To turn on the Tet-

inducible expression, cells were grown in 1 μg/ml Doxycycline (Dox) and nucleofected with sgKAT5:Cas9 complexes to KO endogenous KAT5 (using an sgRNA that spans an exon-intron junction that does not recognize the KAT5 ORF). After a 72 h recovery period, cells plated in

96 well plates at a frequency of ~.25 cells per well in the presence of Dox and allowed to outgrow for two weeks with media changes every three day. Afterwards, several clones that preserved the p27-mVenus reporter were picked and evaluated for growth arrest and p27 reporter induction upon Dox withdrawal. Clone 13, used above, displayed the most uniform Dox+ growth and Dox- growth arrest among ~10 clones evaluated.

**GSC invasion assays.** GSC-0827 C13 cells were cultured with 1 µg/ml Doxycycline or without Doxycycline for 4 days or 7 days. A total of $5 \times 10^3$ cells were seeded in an ultra-low attachment round bottom 96-well plate (Corning, 7007) and incubated 3 days. For the invasion assay, an 8-well chamber slide (LAB-TEK, 154534) was coated with 30 µg/ml Matrigel (Corning, 356231) and incubated at 4 °C overnight, followed by washing with ice-cold PBS. Then, 3–5 spheroids were transferred into 150 µl of 7.8 mg/ml Matrigel-covered wells and incubated at 37 °C for 1 h. After polymerization, 150 µl NSC media were overlaid with or without 2 µg/ml Doxycycline as needed. Phase-contrast images were captured every 24 h for 72 h on a Nikon Eclipse TS100 microscope. The area covered by invading cells was measured using FIJI.

The average cell invasion distance for each spheroid was calculated using the following formula: Average cell migration distance $= \sqrt{End\ migration\ area / \pi} - \sqrt{Start\ migration\ area / \pi}$.

**scRNA-seq analysis.** Single cell RNA-sequencing was performed using 10x Genomics' reagents, instruments, and protocols. Single cell RNA-Seq libraries were prepared using Chromium Single Cell 3′ Reagent Kit. CellRanger[107] (v5.0 from 10× Genomics) was used to align, quantify, and provide basic quality control metrics for the scRNA-seq data. Souporcell[108] was used to deconvolute scRNA-seq data for each GSC cell line. Using Seurat[109] (version 4), the scRNA-seq data was normalized using the SCTransform pipeline and were merged the tumor replicates to build an integrated reference. FindTransferAnchors and MapQuery from Seurat, was used to map the query tumors to the integrated reference. FindAllMarkers was used to find differentially expressed genes for each cluster of each tumor. AddModuleScore from Seurat to calculate the average expression levels of different gene lists of interest for each tumor type. ggplot2 was visualize to make bar plots to visualize the number of genes and cells in each cluster. ccSeurat[109] and ccAF[5] were used to score cell cycle states for each cell. scVelo[110] was used to perform velocity analysis. The ToppCell Atlas[111] was used to perform gene set enrichment analysis on each of the differentially expressed gene list from each cluster.

**CUT&Tag analysis.** CUT&Tag assays were performed using the Epi-Cypher CUTANA CUT&Tag kit (Cat# 14-1101) according to manufacturer's instructions. We used 100,000 cells as input per each sample and proceeded with nuclei extraction before binding to ConA beads. The samples were derived from GSC-0827 C13 orthotopic xenografts that were also used for scRNA-seq analysis in this study. CUT&Tag analysis was assayed for the following: V5 tag (Invitrogen, Cat# R960-25); epigenetic markers: H3K4Me2 (Millipore, Cat# 07-030), H3K27Ac (Millipore, Cat# MABE647) and H3K27Me3 (Cell Signaling Technologies, Cat# 9733).

**Statistics & reproducibility.** Results were statistically evaluated by paired and unpaired t-tests, two-way ANOVA tests, Kolmogorov–Smirnov tests, and linear regression analysis using Prism GraphPad PrismTM software (GraphPad Software Inc., San Diego, CA, USA). Graphs show mean and standard error to the mean (SEM). Values of $p < 0.05$ are considered statistically significant. The use of statistical test, the $p$ value, the numbers of samples and experiments are indicated in the respective figure legends. For sgRNA-seq, statistical significance was tested in use a negative binomial test and Benjamini-Hochberg False

Discovery Rate (FDR) correction using edgeR ($n = 3$ per condition). For scRNA-seq analysis, >1000 cells was used for each scRNA-seq analysis with a per cell sequence read depth of >20,000. Standard Seurat filters were applied requiring that the cells had to have at least 200 features per cell, and transcripts need to be expressed in at least three cells. Each sample was further filtered by requiring the number of UMIs per cell to fall within a range of UMIs and the mitochondrial percentage of genes expressed per cell to fall within a range of percents (as shown in Supplementary Figs. 4, 7). The cells from each sample were then normalized using SCTransform, principal components were calculated, and a UMAP was generated. To identify UMAP gene cluster markers, a Wilcoxon rank sum test and Bonferroni correction were used. To compare cluster-specific gene expression (i.e., differential expression analysis), negative binomial tests (DESeq2) and Benjamini-Hochberg False Discovery Rate (FDR ≤ 0.05) corrections were used. The bioinformatics analysis was processed at least twice, and results were compared to ensure the reproducibility of our findings.

Gene set enrichment analysis utilized Hypergeometric Test and Benjamini-Hochberg False Discovery Rate (FDR) correction. For CUT&RUN and CUT&TAG data sets the following statistical analyses were used: a Poisson Distribution Model and Benjamini-Hochberg False Discovery Rate (FDR) correction for determining significantly enriched genomic sites (MACS analysis); a negative binomial test and Benjamini-Hochberg False Discovery Rate (FDR) correction for identifying differentially enriched sites between control and experimental data sets (DiffBind); and a Hypergeometric Test and Benjamini-Hochberg False Discovery Rate (FDR) for identifying enrich transcription factor motifs (HOMER). Only mouse tumor experiments were randomized. Investigators were not blinded to allocation during experiments and outcome assessment.

## Reporting summary

Further information on research design is available in the Nature Portfolio Reporting Summary linked to this article.

## Data availability

The sgRNA-seq and scRNA-seq data files are publicly available on the GEO database at GSE198524. https://www.ncbi.nlm.nih.gov/geo/query/acc.cgi?acc=GSE198524. All other data associated with this study are present in supplementary materials and tables. Source data are provided with this paper as a Source Data file. Source data are provided with this paper.

## Code availability

The code used to process and analyze the data is available at https://github.com/sonali bioc/GSC_scRNASeq_KAT5paper; https://github.com/plaisier-lab/KAT5paper.

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

## Acknowledgements

We thank members of the Holland, Paddison, Patel, Plaisier, and Tsukiyama labs for helpful discussions, Dr. Atsushi Miyawaki for providing reagents, and Pam Lindberg and An Tyrrell for administrative support. This work was supported by the following grants: Interdisciplinary Training in Cancer Fellowship NCI T32CA080416 (P.H.); Fred Hutch pilot award (A.P., P.P.); NCI/NIH (R01CA190957; R01CA295090; P30CA15704) (P.P.); (5R21CA232244) (C.P.); NINDS/NIH (R01NS119650) (A.P., C.P., P.P.); NIGMS (R35GM139429)(T. T.); Burroughs Wellcome Career Award for Medical Scientists (A.P.); and a grant from the Kuni Foundation (A.P.). Preclinical Imaging Core, Fred Hutchinson Cancer Center support: P30 CA015704 (RRID:SCR_022616); 3 T/7 T MRI SIG: NIH S10 OD26919.

## Author contributions

Project conception and design was carried out by P.P., A.P., C.P., T.T., A.M., and H.F. Experiments and data analysis were performed by A.M., H.F., J.B., K.M., C.C., P.H., M.L., and W.J. Critical reagents were generated by J.D., M.K., P.C., and L.C.; single cell bioinformatical data analysis and statistics were performed by S.A., S.O., K.J., B.A. and C.P. with input from A.P., C.P., and P.P.; P.P., A.P., C.P. and A.M. wrote the manuscript with input from other authors.

## Competing interests

The authors declare no competing interests.
