## [Transparent Peer Review file · Nature Communications]

KAT5 regulates neurodevelopmental states associated with G0-like populations in glioblastoma

Corresponding Author: Dr Patrick Paddison

Version 0:

Reviewer comments:

Reviewer #1

(Remarks to the Author)

The authors of this work have embarked on an intriguing study investigating G0 cell populations in GBM, an area typically less explored. They have employed a broad range of experimental models and high-dimensional data acquisition to identify KAT5 as a crucial epigenetic regulator of the G0 state. However, the subsequent experiments scrutinizing the impact of KAT5 loss failed to yield any novel, clinically relevant insights, and did not adequately address the compelling questions surrounding G0 cell populations. Major concerns include:

The use of an orthotopic GBM model is problematic without validation through exploration and distribution on real-world patient data. Available reference datasets from hundreds of patients, including spatial data, could be instrumental in contextualizing the findings from the orthotopic GBM model within accepted classification systems and human GBM data. The lack of an integrative presentation of the data raises several questions. For instance, given that hypoxia-affected cells are significantly enriched in the p27hi group, how does the tumor's spatial organization, from necrosis to infiltration, and the associated variance of metabolic conditions, influence the distribution of G0 and proliferative cells? Recent research has indicated that hypoxia niches almost completely lack G2/M cells due to metabolic induced inhibition of the cell cycle (S-Phase arrest).

In the second part, the experiment examining the influence of KAT5 on tumor growth seemed somewhat anticipated, as a lower frequency of mitosis will inevitably result in a less proliferative tumor. Although this finding may serve as validation, it does not enhance our understanding of these cells' role. Considering the authors have acknowledged issues such as resistance of GSC and neuronal interaction, it would have been rewarding to see experiments designed to clarify the role of the identified G0 stem-cell like population. The mouse and TCGA analyses were unconvincing in demonstrating any clinical association.

Furthermore, the paper did not discuss functional networks within the tumor. As a feature of AC-like cells, does the KAT5 KO alter morphology or cell-cell signaling? What about the cells' migration and interaction with neuronal cells? Are there different responses to radiotherapy or chemotherapy? While the data suggest that a high proportion of G0 cells is beneficial, does this mean patients are less responsive to chemotherapy? What external events might awaken these cells from their dormancy, given that all gliomas eventually recur?

The final experiment, attempting to validate the activity of KAT5 in patient samples, conflates two entirely different cancer entities. Recent research, culminating in the WHO classification, distinguishes GBM and IDH mutated tumors. There is no transition from LGG (IDH mutated) to HGG (GBM). IDH mutated Astrocytoma Grade 4, the malignant variant of an IDH mutated tumor, bears no relation to GBM.

IDH mutated and GBM WT samples cannot be directly compared, especially concerning epigenetic modifications. Given that the exploratory phase of the study commenced with WT tumors, the authors should remain consistent and stick to WT tumors. Otherwise, all experiments would need to be redone using an orthotopic IDH mutated model.

Minor points include:

Figure 1 would benefit from a bar plot representation of the relative difference in cell abundance between the p27hi and low

groups, improving the visualization of differences between the total and p27hi group.

It would be beneficial to explore the effect of KAT5 KO using in-silico perturbation on the in-vivo dataset, such as CellOracle. Combined with velocity, this would offer valuable insights into the transformation of the initial cell population, which differs from the cell culture screening.

In terms of KAT5 expression in a human dataset, how is the expression associated with the known Neftel states? What is the spatial variation in KATA5 expression?

For Figures 6f and g, it is unclear why DOX + is presented first. Conventionally, the control would be expected in the first line, as shown in previous figures.

Reviewer #2

(Remarks to the Author)

In this large data study, Mihalas et al., examine the ingress/egress regulation of G0-like glioma cells and its potential role in “tumor growth and recurrence”. While the strategy and effort are impressive to establish the function of KAT5 in cell proliferation, epigenetics, and protein synthesis, the importance of KAT5 in tumor maintenance and recurrence is unclear. The authors started with a single human patient-derived GBM cell line (GSC-0827) conjugated with a reporter for G0 (p27-mVenus). P27 expression is selected on the basis of its inverse correlation with cyclin protein expression. scRNA-seq of the whole tumor revealed ~34% of heterogeneous cycling cells, consisting of S/G2/M cells. Reporter p27hi cells are underrepresented in the cell clusters expressing active cycling genes but present (mostly in the 10-15% range) in all in the non-proliferative cell clusters. One cluster Cell (cluster 1) shows 25% reporter expression and is designated the G0/G1 cluster. Combined with differential gene expression analysis, gene set enrichment analysis, and a cell cycle classifier tool, they further delineate G0/G1 cells as “oligodendrocyte progenitor cells (cluster 6) (OPC), radial glial cells/astrocytes (RG) (cluster 8), cycling neural progenitors (clusters 1/3), and hypoxia (cluster 4)”. Are any of these states considered quiescent? Stem cells are usually thought of being quiescent for example.

The authors then turned to whole genome CRISPR screen to identify genes egressing cells from G0-like states. It is confusing to me how cells proliferating in culture can be thought of as being in a G0-like state. The 75 genes enriched in the p27hiCDT1+ cells contain the ribosome assembly and ribosome protein coding genes (e.g., RPL5, RPS16,) and the Tip60/NuA4 lysine acetyltransferase complex including KAT5. They then validated the role of KAT5 in ingress/egress of G0-like states in GSCs with knock-outs and an inducible KAT5 rescue construct. KAT5off tumor cells enrich G0-like OPC by ~16-fold and RG-like clusters by 6-fold. Epigenetic analyses disclose super-enhancers associated with KAT5 states. It seems that the readout for these functional assays are proliferation rather than in vitro and in vivo assays illustrate the function of KAT5 in glioma cell proliferation and not necessarily in a G0 egress. Finally, genes inhibited by KAT5 are correlated with low grade gliomas. Though KAT5 appears to be critical for glioma cell proliferation, “GSC cells, none-the-less, retain the same cellular mass as control cells, even after 14 days of KAT5 off”. Considering the fact that the standard care of GBM patients targeting proliferative cells with chemoradiotherapy failed to prevent tumor recurrence, it is not likely that KAT5 inhibition will achieve the goal. It might be helpful for the authors to incorporate temozolomide treatment in KAT5on and KAT5off cells if they really want to address tumor recurrence.

Major concerns:

- This manuscript relies upon the study of a single patient derived tumor cell line. Therefore, how generalizable the result may be come into question.
- The premise of the “functional states” of tumor cells is computational with the one exception of p27-versus reporter. A variety of gene signatures are used but no experimental support for how these cells move, other than difficult to interpret velocity plots.
- The rationale behind using cell culture assays and screens to identify cells in a G0 (quiescent state) is not made clear and would require a clear defense.
- Kat5 appears to be a very interesting gene but from the data presented it is difficult to conclude beyond that its absence reduces cell proliferation (EdU incorporation) and that many genes appear to be regulated by it.

Additional Comments:

1. FIG. 1b, cluster 1 is not specified. Clusters 3 and 7 are annotated as cycling neural progenitors, which does not match the description in the draft: “cycling neural progenitors (clusters 1/3)”. Please clarify.
2. ToppCell Fig 1F shows highest score for cluster 1 in stem cells/neuroepithelial cells/cycling progenitors/and cycling cells. But all the stem cell markers are higher 0, 5, and 3 which are also high cycling. Cluster 1 which has the highest p27 score has a cycling signature. Please explain.
3. Fig. 2, I am not sure how the mRNA velocity plots help.
4. Fig. 3h, the third panel (cell cycle) does not match with the legend (MES:mecsenchymal). Please clarify.
5. Fig. 6e, It would be helpful to have MRI images side by side from the different conditions, or simply remove the panel if not interesting.

6. Fig. 6h, it seems glioma cells can still grow even without KAT5 (Dox0), though much slower. Is this why they “retain the same cellular mass”? How about the survival curves? Do they show difference between the KAT5on and KAT5off cells?

7. The authors identified cluster 8 cells as radial glial cells expressing genes in “neurogenesis or glioma biology by promoting stemness, proliferation, invasive behavior, key signaling pathway (e.g., Notch), or survival in GSCs and/or NSCs”. Is it possible these are stem-like cells? This might explain why the KAT5off cells can still re-propagate tumor mass.

Reviewer #3

(Remarks to the Author)

The manuscript by Mihalas et al. is an interesting work identifying Kat5 as a regulator of glial precursor genes and cell cycle. The paper is very well written and data clearly presented. Some points need to be considered:

1. Figure 1: The data is based on one GSC model. Is this reproducible in a second model? Will the same clusters be detectable in a second model? What about the distribution of p27^{hi} and p27^{low} cells in a second model? It is difficult to base the results on one model. It is well known that GBM show high intertumoral heterogeneity.
2. If p27^{hi} cells do not reenter the cell cycle, are the cells tumorigenic in vivo? And if yes, tumor growth should be delayed. It is highly recommended to compare growth of p27^{hi} to p27^{lo} tumors in vivo. This would bolster the authors findings.
3. Figure 5: SOX genes activated in Kat5 off tumors are also involved in cell cycle progression, how does these data fit into the authors concept? This could also be discussed more broadly how the activation of certain genes in Kat5 off tumors fit into the concept of G0 state as some of these genes are known to be expressed at high levels in proliferating brain tumors including olig2.
4. Figure 6 would benefit from a survival study (comparing Dox0 and Dox2000); MRI volumes look convincing; however, some invasive tumor portions might not be detectable by conventional MRI. In addition the authors could use a simple model of KAt5 ko versus ctrl in a second model to confirm impact on survival/tumorigenicity
5. The surprising observation of figure 6j,k should be confirmed with a second cell line and another invasion assay.
6. How is Kat5 expression in low grade compared to high grade glioma? The authors should investigate this using available databases. High grade tumors should have higher levels of Kat5 expression according to the data of their C13 animal model.

Version 1:

Reviewer comments:

Reviewer #1

(Remarks to the Author)

The authors have refocused the manuscript on KAT5, addressing most of the previously raised concerns. The revisions have significantly improved the manuscript, and the majority of issues have been resolved.

However, a few minor points remain:

The response: “We observed significantly higher expression in grade 2 and 3 gliomas and also IDH1/2 mutant gliomas,” is unclear. Could the authors clarify which grade 2 and 3 gliomas are not IDH-mutated? Furthermore, do the results indicate any differences between astrocytomas and oligodendrogliomas within these grades?

The phrase “not shown” should be removed from the manuscript, as it does not provide helpful context.

Overall, the manuscript is much improved and now provides valuable insights into the role of KAT5.

Reviewer #3

(Remarks to the Author)

The authors provided a largely extended manuscript with a lot of new data. The referee thinks that there is enough experimental evidence showing that KAT5 increases the proliferative state in GB cells. However, the data also indicate that in each GB there are fractions with lower and higher KAT5 activity which correlates to different GB states such as mesenchymal versus OPC/RG-like states in C13 tumors. Was the mesenchymal state found in C13 KAT5 on a significant finding? This was not so clear from the manuscript and from main Figure 5D. At least, KAT5 activity seems to provide a switch between GB states based on the authors data. This finding of switching between states should be shown in a broader picture as it is highly relevant for GB. The switching of states could be correlated to H4-Ac and OPP. So if the authors could sort the two different states H4-Ac and OPP high versus low from GB patient samples and provide RNAseq clustering would be beneficial for the paper (Figure 7e-i).

The sensitivity to SOC seems not to be very different between KAT5 on vs off cells. The authors could compare the gain in survival benefit of SOC compared to control in both KAT5 off and on (Figure 6h).

The regulation of genes between KAT5 on and off (Figure 5i) further supports the notion of switching between GB states, as

a number of genes found to be upregulated in KAT5 off are also found in areas of GB samples. This underscores that most likely KAT5 regulates the switching between GB states and should be further supported by the above-mentioned experiment.

Additional comments:

1. Figure 1g: The referee could not find K8 and K16 levels in the blot as stated in the manuscript.
2. Invasion assay into Matrigel is a bit artificial. The authors could at least use another method to study invasion (in 2 cell lines), the best would be if they have brain organoids or cortical slice cultures at hand, which are the optimal settings to measure invasion
3. Figure 8a: What would happen with NSC-KAT5? What is the status of KAT5 activation in NSC? Would NSC-KAT5 have upregulation of MYC?
4. The drug HHT most likely does not cross the BBB as it is more than 500 in molecular weight. This is a major drawback using such a drug in the context of GB, this should be discussed.
5. A recent paper showed involvement of KAT5 in cell proliferation and growth in GB: PMID: 38172338. This paper should be discussed, it is supportive of the authors data. Other GB and KAT5 papers that could be discussed: PMID: 34077757, PMID: 29677490

Version 2:

Reviewer comments:

Reviewer #3

(Remarks to the Author)

all comments sufficiently addressed.

REVIEWER COMMENTS

General comments to reviewers for resubmission.

First, we would like to thank the reviewers for their thoughtful critiques of the previous submission. We have spent >1 year completing experiments to address most of the reviewers concerns and have now resubmitted, we feel, a much-improved manuscript. However, we have altered the narrative to focus more on KAT5 function.

After completing much of the work, we realized we have an abundance of data regarding G0 states which is not directly linked to KAT5 function. At the same time, we have also generated multiple new data sets around KAT5 function which we have incorporated into this revised manuscript. These include: analysis of KAT5 induced cell states in nine independent GSC patient isolates using scRNA-seq (new Fig. 3), new scRNA-seq tumor references of 2 different GSC-derived tumors which we leverage for analysis of KAT5 cells states (new Fig 4), KAT5 chromatin binding data from GSC-derived tumors (new Fig. 5), as well as further analysis of factors driving KAT5 activity in high grade glioma (new Fig 8) and identification of candidate therapeutic approach in KAT5^{hi} tumors (new Fig 8).

As a result, we are preparing a separate manuscript that focuses on G0 states in GBM that will address a portion of the comments from the previously submission as noted below (including characterization of p27^{hi} and label retaining populations). However, we feel that the KAT5 story is strong enough to stand on its own and still shares much of the data and concepts of the previous manuscript.

Reviewer #1 (Remarks to the Author): expertise in glioblastoma scRNA-seq
Major concerns :

The use of an orthotopic GBM model is problematic without validation through exploration and distribution on real-world patient data. Available reference datasets from hundreds of patients, including spatial data, could be instrumental in contextualizing the findings from the orthotopic GBM model within accepted classification systems and human GBM data. The lack of an integrative presentation of the data raises several questions. For instance, given that hypoxia-affected cells are significantly enriched in the p27^{hi} group, how does the tumor's spatial organization, from necrosis to infiltration, and the associated variance of metabolic conditions, influence the distribution of G0 and proliferative cells?

Response: *We have performed scRNA-seq on 2 additional GSC tumors and spatial CUT&TAG on two of these as well as integrated the data with existing spatial-seq data sets from GBM tumor isolates. However, as noted above we realized that this is simply too much data for the KAT5 manuscript. We are opting to include this data in a separate manuscript which focuses on defining G0 states in GSC tumors. However, for the current manuscript, we have included use of our computational classifier for identifying candidate G0-like states in GBM samples which matches well with states induced by KAT5 inhibition. We have also used accepted classification systems (i.e., gene expression modules) for examination of our GSC-derived tumor subpopulations.*

In the second part, the experiment examining the influence of KAT5 on tumor growth seemed somewhat anticipated, as a lower frequency of mitosis will inevitably result in a less proliferative tumor. Although this finding may serve as validation, it does not enhance our understanding of these cells' role. Considering the authors have acknowledged issues such as resistance of GSC and neuronal interaction, it would have been rewarding to see experiments designed to clarify the role of the identified G0 stem-cell like population. The mouse and TCGA analyses were unconvincing in demonstrating any clinical association.

Furthermore, the paper did not discuss functional networks within the tumor. As a feature of AC-like cells, does the KAT5 KO alter morphology or cell-cell signaling? What about the cells' migration and interaction with neuronal cells? Are there different responses to radiotherapy or

chemotherapy? While the data suggest that a high proportion of G0 cells is beneficial, does this mean patients are less responsive to chemotherapy? What external events might awaken these cells from their dormancy, given that all gliomas eventually recur?

Response: *We completed comparative analysis of KAT5 inactivation in 9 separate GSC isolates using scRNA-seq as readout and show that KAT5 loss induces novel cell states in vitro that are associated with quiescent-like states in tumor models from these tumors (new Figs 3 and 4). We further show that KAT5 activity promotes GSC self-renewal through coordinately regulating E2F- and MYC-transcriptional networks with ribosome assembly and protein translation. Inhibiting KAT5 causes simultaneous loss of transcription of key E2F- and MYC- targets while attenuating protein synthesis. As this occurs, GSCs and GSC-derived tumor cells transition into G0-like states with neurodevelopmental features commonly observed in OPC and RG cells, subpopulations of GBM tumors, and the computationally defined Neural G0 state (e.g., PTN/PTPRZ1). We further show that KAT5 inactivation attenuates tumor growth and has a significant survival benefit both with or without standard of care. Finally, we show that KAT5hi cells are sensitive to the clinically approved protein translation inhibitor homoharringtonine. Both of the KAT5hi state and HHT sensitivity could be induced by expression of MYC in hNSCs. We provide a general model for these findings in Fig 8c. In general, the data are consistent with the notion that down regulating KAT5 in high grade tumors would promote more indolent tumor growth which has survival benefits.*

The final experiment, attempting to validate the activity of KAT5 in patient samples, conflates two entirely different cancer entities. Recent research, culminating in the WHO classification, distinguishes GBM and IDH mutated tumors. There is no transition from LGG (IDH mutated) to HGG (GBM). IDH mutated Astrocytoma Grade 4, the malignant variant of an IDH mutated tumor, bears no relation to GBM.

Response: *We did not mean to conflate the IDH-wt with IDH-mut tumors. We have broken these tumors up and redid the analysis (Supplementary Fig. 14). We also examined whether genes upregulated after KAT5 loss differ significantly in expression in clinical glioma tumor surgical samples by IDH1/2 mutation status and/or grade. We observed significantly higher expression in grade 2 and 3 gliomas and also IDH1/2 mutant gliomas. They also significantly predicted survival in IDH1/2 mutant gliomas (pval.=.006) and show a strong tendency for IDHwt gliomas (pval.=0.099). However, KAT5 expression on its own failed to predict survival in IDH1/2 mutant or wt gliomas.*

For KAT5 activity assays we wished to illustrate multiple points – mainly that both KAT5 activity and protein translation are higher in HGG IDH wt and mut tumors than LGG and that KAT5 activity directly correlates with protein translation in tumors regardless of grade or IDH status and that KAT5 activity is dynamic. We now show that KAT5 activity is associated with sensitivity to HHT, which underscores these distinctions further. We have tried to further clarify these points.

Reviewer #2 (Remarks to the Author): expertise in glioma quiescence and stem cells

Major concerns:

- This manuscript relies upon the study of a single patient derived tumor cell line. Therefore, how generalizable the result may be come into question.

Response: *we have added 8 additional GSC models with in vitro scRNA-seq data and another in vivo tumor reference (new Figs 3 and 4).*

- The premise of the “functional states” of tumor cells is computational with the one exception of p27- versus reporter. A variety of gene signatures are used but no experimental support for how these cells move, other than difficult to interpret velocity plots.

Response: we have created an experimental paradigm and results for this but decided to include it and the p27hi single cell data in the sister G0 manuscript – thank you for this suggestion.

- The rationale behind using cell culture assays and screens to identify cells in a G0 (quiescent state) is not made clear and would require a clear defense.

Response: We have attempted to clarify the rationale of the screen in first part of the results section

- Kat5 appears to be a very interesting gene but from the data presented it is difficult to conclude beyond that its absence reduces cell proliferation (EdU incorporation) and that many genes appear to be regulated by it.

Response: As noted above we have better resolved the mechanism of KAT5 function our experimental models. We find that inhibiting KAT5 causes simultaneous loss of transcription of key E2F- and MYC- targets while attenuating protein synthesis. As this occurs, GSCs and GSC-derived tumor cells transition into G0-like states with neurodevelopmental features commonly observed in OPC and RG cells, subpopulations of GBM tumors, and the computationally defined Neural G0 state (e.g., PTN/PTPRZ1).

- It might be helpful for the authors to incorporate temozolomide treatment in KAT5on and KAT5off cells if they really want to address tumor recurrence.

Response: We show that KAT5 inactivation attenuates tumor growth and has a significant survival benefit both with or without standard of care (Fig 6). We also find that that KAT5hi cells are sensitivity to the clinically approved protein translation inhibitor homoharringtonine (Fig 8). Both of the KAT5hi state and HHT sensitivity could be induced by expression of MYC in hNSCs (Fig 8)

Reviewer #3 (Remarks to the Author): expertise in glioblastoma mouse models

The manuscript by Mihalas et al. is an interesting work identifying Kat5 as a regulator of glial precursor genes and cell cycle. The paper is very well written and data clearly presented. Some points need to be considered:

1. Figure 1: The data is based on one GSC model. Is this reproducible in a second model? Will the same clusters be detectable in a second model?

Response: we have added 8 additional GSC models with in vitro scRNA-seq data and another in vivo tumor reference (new Figs 3 and 4).

2. If p27hi cells do not reenter the cell cycle, are the cells tumorigenic in vivo? And if yes, tumor growth should be delayed. It is highly recommended to compare growth of p27hi to p27lo tumors in vivo. This would bolster the authors findings.

Response: we have performed this experiment --- it worked! we are included in the sister manuscript as noted above.

3. Figure 5:SOX genes activated in Kat5 off tumors are also involved in cell cycle progression, how does these data fit into the authors concept? This could also be discussed more broadly how the activation of certain genes in Kat5 off tumors fit into the concept of G0 state as some of these genes are known to be expressed at high levels in proliferating brain tumors including olig2.

Response: *In the new submission we find that inhibiting KAT5 causes simultaneous loss of transcription of key E2F- and MYC- targets while attenuating protein synthesis. As this occurs, GSCs and GSC-derived tumor cells transition into G0-like states with neurodevelopmental features commonly observed in OPC and RG cells, subpopulations of GBM tumors, and the computationally defined Neural G0 state (e.g., PTN/PTPRZ1). However, this comment raises a few important points about how we interpret these results. However, one way we can formulate an answer from our experimental data from GSC-0827 Clone 13 Dox-KAT5 cells. These cells are derived from a clone which are KO for endogenous KAT5 but are complemented with tetO7-KAT5 (Dox+). The tumor arising from this single clone shares the cell states of the GSC-0827 parental tumor and aggressive tumor formation. However, upon Dox withdrawal the tumor becomes enriched for OPC and RG-like cell populations and is higher for p27+, lower for EdU+, and have significant survival benefits with and without SOC. I'm wondering whether the reviewer and / or editor would like us to discuss this point in the results and discussion. We appreciate the general point regarding OLIG2+ proliferative GBM populations, however, our experimental results suggest that G0 populations also have enrichment for this marker in the experimental models we are using. We do have a two computational papers that seem to suggest this as well – our next G0 state paper will also suggest this – but we fully appreciate that expressions of individual markers might vary in tumors.*

4. Figure 6 would benefit from a survival study (comparing Dox0 and Dox2000); MRI volumes look convincing; however, some invasive tumor portions might not be detectable by conventional MRI. In addition the authors could use a simple model of KAT5 ko versus ctrl in a second model to confirm impact on survival/tumorigenicity.

Response: *we have added 8 additional GSC isolates and another tumor for our KAT5 KO comparisons and also incorporated SOC in vivo and identified a candidate KAT5hi therapeutic agent (though it requires further study)(i.e., HHT)(Fig. 8).*

5. The surprising observation of figure 6j,k should be confirmed with a second cell line and another invasion assay.

Response: *we performed the invasion assay in 5 additional GSC isolates.*

6. How is Kat5 expression in low grade compared to high grade glioma? The authors should investigate this using available databases. High grade tumors should have higher levels of Kat5 expression according to the data of their C13 animal model.

Response: *we confirmed that KAT5 expression itself is not associated with survival benefit in LGG or HGG tumors and show in Supplementary Fig. 16 KAT5 expression in all available CNS tumors and control tissues. We also performed KAT5 activity assays in NSCs which all turned out to be more interesting than expected (Fig 8).*

Reviewer #1 (Remarks to the Author):

The response: "We observed significantly higher expression in grade 2 and 3 gliomas and also IDH1/2 mutant gliomas," is unclear. Could the authors clarify which grade 2 and 3 gliomas are not IDH-mutated? Furthermore, do the results indicate any differences between astrocytomas and oligodendrogliomas within these grades?

Response: Thank you for pointing out this poorly constructed sentence.

We have changed it to: "We observed significantly higher expression in grade 2 and 3 IDH1/2 wt gliomas compared to grade 4 IDH1/2 wt tumors, and also higher expression in IDH1/2 mutant grade 2 and 3 gliomas compared to IDH1/2 wt grade 2 and 3 tumors (**Supplementary Fig. 14**). In addition, we also find significantly higher expression in grade 2 versus grade 3 tumors for astrocytomas, oligoastrocytomas, and oligodendrogliomas; and higher expression in each of these LLG tumor types when compared to GBM regardless of grade (**Supplementary Fig. 14**)." (Lines 372-379)

The phrase "not shown" should be removed from the manuscript, as it does not provide helpful context.

Response: We have removed the phrase.

Reviewer #3 (Remarks to the Author):

The authors provided a largely extended manuscript with a lot of new data. The referee thinks that there is enough experimental evidence showing that KAT5 increases the proliferative state in GB cells.

A) Was the mesenchymal state found in C13 KAT5 on a significant finding? This was not so clear from the manuscript and from main Figure 5D. At least, KAT5 activity seems to provide a switch between GB states based on the authors data. This finding of switching between states should be shown in a broader picture as it is highly relevant for GB. The switching of states could be correlated to H4-Ac and OPP. So if the authors could sort the two different states H4-Ac and OPP high versus low from GB patient samples and provide RNAseq clustering would be beneficial for the paper (Figure 7e-i).

Response: Thank you for making this point. We have tried these experiments using QC measure for single cell nuc-seq and whole cell-seq. These experiments involve intracellular flow analysis of both H4-Ac and OPP in patient isolates. Unfortunately, we have been unable to achieve reasonable RIN (RNA integrity number) scores and pilot single cell sequencing suggested that the approach requires additional optimization. Thus, at this time, these experiments are not feasible, but we hope to achieve success in the future. What we were able to show that KAT5 activity, as read out by H4-Ac, is dynamic in tumors and is directly correlated with OPP state. By manipulating KAT5 activity in vitro and in xenografts directly in patient isolates, we are able to identify the gene expression clusters associated with KAT5_{low} state. Regarding the mesenchymal state, we have performed statistical test of mesenchymal genes down regulated in KAT5_{off} vs. KAT5_{on} states (Fig 5e). We have added the word "significant" to the sentence describing Fig 5e (Lines 286-288).

B) The sensitivity to SOC seems not to be very different between KAT5 on vs off cells. The authors could compare the gain in survival benefit of SOC compared to control in both KAT5 off and on (Figure 6h).

Response: We have added a sentence detailing average days gained in mouse survival from KAT5 on vs. off and with and without radiation (days gained: Dox- = 17.75; SOC/Dox+: 17.97; and SOC/Dox- =39.55) in legend of Figure 6, lines 25-27. The statistical comparisons for gain in survival benefit are now shown in Suppl Fig 13a.

Additional comments:

1. Figure 1g: The referee could not find K8 and K16 levels in the blot as stated in the manuscript.

Response: Thank you for pointing this out. This was an omission for this figure -- we have added those data to Fig 1g.

2. Invasion assay into Matrigel is a bit artificial. The authors could at least use another method to study invasion (in 2 cell lines), the best would be if they have brain organoids or cortical slice cultures at hand, which are the optimal settings to measure invasion.

Response: We agree that matrigel invasion assays do not fully capture the complexity of tumor invasion in vivo. However, they are able to capture cell-ECM interactions and proteolytic invasion. We performed our in vitro matrigel invasion assays with 6 separate GSC isolates, with 5 showing significant reductions in invasiveness after loss of KAT5 activity (Fig.6 i&j; Suppl Fig. 13). We are continuing to work on this phenotype, ultimately with xenograft models, and are adding this data to a grant application for additional funding to pursue this phenotype. We feel this phenotype is not critical to the central message of this manuscript but does add additional interesting perspective on KAT5 function. We could certainly remove this data. However, it was done in collaboration with Dr. Jon Cooper's group who is expert on cell migration and invasion and this is a commonly used in vitro invasion assay for tumor cells.

3. Figure 8a: What would happen with NSC-KAT5? What is the status of KAT5 activation in NSC? Would NSC-KAT5 have upregulation of MYC?

Response: Thank you for this suggestion. We, in general, focus on requirement studies using knockdown or knockout as these are more interpretable. In the case of MYC, its ectopic expression is well known to be sufficient to induce cells to proliferate and we had previously shown this to be the case for human NSCs. However, since KAT5 is part of a multi-subunit complex and has multiple regulatory activities, interpreting overexpression experiments is difficult and so we chose to limit KAT5 ectopic expression studies to those in which endogenous KAT5 knockout was complemented. Nonetheless, we do track KAT5 activity in NSC cells shown in Figure 8a and clearly show that MYC is necessary to elevate KAT5 activity levels and translation rates in NSCs.

4. The drug HHT most likely does not cross the BBB as it is more than 500 in molecular weight. This is a major drawback using such a drug in the context of GB, this should be discussed.

Response: This is a salient point, we have added this to the discussion, as well as the following reference (PMID: 3598625), which shows that HHT can traverse the BBB.

5. A recent paper showed involvement of KAT5 in cell proliferation and growth in GB: PMID: 38172338. This paper should be discussed, it is supportive of the authors data. Other GB and KAT5 papers that could be discussed: PMID: 34077757, PMID: 29677490

Response: Thank you for bringing these references to our attention. We have added them to the discussion (lines 529-533).

Regarding Reviewer #2's report:

I consider most of the author's responses as sufficient except:

- The premise of the "functional states" of tumor cells is computational with the one exception of p27-vensus reporter. A variety of gene signatures are used but no experimental support for how these cells move, other than difficult to interpret velocity plots.

Response: we have created an experimental paradigm and results for this but decided to include it and the p27hi single cell data in the sister G0 manuscript – thank you for this suggestion.

I am not an expert on scRNA, and I am also not sure which type of experiments the referee meant. But I also think that the author's response should be more elaborated. If they have new data concerning this point, they

should provide a rational argument why they do not want to include them in this manuscript but use it for another manuscript.

Response: We apologize for the terse response to this point. To obtain support for how these cells "move", we have used label retaining experiments to better define longer term quiescent populations in tumors using a Dox inducible label retention reporter in xenograft tumors. These experiments and the single cell data associated with them, including the p27hi scRNA-seq, created a significant amount of additional data that was not directly relevant to KAT5 activity, and we felt better suited for a second manuscript, titled "Characterization of quiescent subpopulations and proliferative compartments in glioblastoma". In this manuscript, we attempt to provide additional detail on feature and properties of quiescent GBM cells. We spend a significant portion of this manuscript computationally comparing quiescent and proliferative compartments of tumors and do not directly delve into KAT5 activity or function.

- The rationale behind using cell culture assays and screens to identify cells in a G0 (quiescent state) is not made clear and would require a clear defense.

Response: We have attempted to clarify the rationale of the screen in first part of the results section

The authors should give a detailed explanation and not just refer to the paper. As I see it, the authors provide a strong argument for in vitro screen as they can recapitulate their findings in vivo. So, I am not in general concerned about this point. But they might make it clearer in the manuscript and also discuss it, because it is usually a concern working on cell cycle in vitro, the cells are much more proliferative compared to in vivo situation, this is a well-known phenomenon.

Response: We apologize for being terse again here. We used patient-derived GBM stem-like cells (GSCs) to identify genes with roles in regulating G0-like states. GSCs are isolated and cultured in serum-free conditions directly from GBM tumors that allow retention of the development potential, gene expression patterns, and genetic alterations found in the patient's tumor. In these conditions in vitro, GSCs typically have shorter overall transit times between mitosis and the next S-phase (i.e., G0/G1) when compared to hNSCs. G0/G1 is ~12hrs for cultured GSCs versus ~33 hrs for hNSCs. This large difference is due to the additional time cultured NSCs spend in a transient G0 state, which is opposed by oncogenic lesions in GBM cells that increase the likelihood of cell cycle entry (e.g., Rb-axis alterations). Because these differences are readily observable in vitro, we sought to identify genes which when inhibited could trigger a pronounced G0-like state in GSCs.

We have amended the text of the beginning results section to emphasize the rationale for in vitro screens.